# Spatiotemporal control of coacervate formation within liposomes

Siddharth Deshpande [1], Frank Brandenburg[1,2], Anson Lau [1,2], Mart G.F. Last[1], Willem Kasper Spoelstra[1], Louis Reese [1], Sreekar Wunnava [1], Marileen Dogterom[1] & Cees Dekker [1]

Liquid-liquid phase separation (LLPS), especially coacervation, plays a crucial role in cell biology, as it forms numerous membraneless organelles in cells. Coacervates play an indispensable role in regulating intracellular biochemistry, and their dysfunction is associated with several diseases. Understanding of the LLPS dynamics would greatly benefit from controlled in vitro assays that mimic cells. Here, we use a microfluidics-based methodology to form coacervates inside cell-sized (~10 μm) liposomes, allowing control over the dynamics. Protein-pore-mediated permeation of small molecules into liposomes triggers LLPS passively or via active mechanisms like enzymatic polymerization of nucleic acids. We demonstrate sequestration of proteins (FtsZ) and supramolecular assemblies (lipid vesicles), as well as the possibility to host metabolic reactions ($\beta$-galactosidase activity) inside coacervates. This coacervate-in-liposome platform provides a versatile tool to understand intracellular phase behavior, and these hybrid systems will allow engineering complex pathways to reconstitute cellular functions and facilitate bottom-up creation of synthetic cells.

[1] Department of Bionanoscience, Kavli Institute of Nanoscience Delft, Delft University of Technology, Van der Maasweg 9, 2629 HZ Delft, The Netherlands. [2] These authors contributed equally: Frank Brandenburg, Anson Lau. Correspondence and requests for materials should be addressed to C.D. (email: c.dekker@tudelft.nl)

Recently, coacervates have aroused enormous interest as membraneless organelles. They emanate from liquid–liquid phase separation (LLPS), primarily of charged polymers (such as proteins and nucleic acids), as a result of favorable attractive interactions over a homogeneous mixture[1,2]. These organelles are ubiquitous inside eukaryotic cells, both in the nucleus as well as in the cytoplasm, a few prominent examples being the nucleolus, germ granules, and stress granules[2–5]. They play a crucial role in intracellular dynamics, regulation, organization and homeostasis, and are associated with several protein aggregation diseases including neurodegenerative disorders such as Alzheimer's disease and amyotrophic lateral sclerosis (ALS)[2,6–9]. One of the most commonly encountered types of condensates are complex coacervates, which form through the electrostatic interaction between oppositely charged polyelectrolytes. Biomolecules can specifically partition and concentrate inside or at the interface of these compartments, depending on molecular interactions and properties such as solubility, hydrophobic stabilization, and electrostatic complementarity[10–13]. Please note that, since the precise terminology in this emerging field is still being developed, we will, in the current paper, use the terms coacervation/condensation interchangeably to refer to the process of LLPS, while the terms coacervate/condensate indicate the condensed reaction products resulting from the process.

The presence of membraneless compartments in cells have sprouted numerous in vitro studies to further understand their properties and dynamics. For example, enzymatic reaction rates have been observed to strongly increase within coacervates, as recently shown for transcription and translation[14]. Additionally, regulation in the form of initiation and dissolution of coacervation has been demonstrated through externally driven enzymatic reactions[15,16]. However, capturing the onset of coacervation and following the subsequent dynamics (nucleation, growth, dissolution, associated chemical reactions, etc.) with high spatiotemporal resolution has found to be challenging in conventional bulk experiments. Indeed, the possibility to induce and limit the coacervation process within a defined volume through external control is highly desirable for in vitro studies.

Here, we explore the use of liposomes as controllable containers for in vitro studies of coacervation. Liposomes are compartments consisting of selectively permeable phospholipid bilayers, similar to those found in living cells in the form of cell membrane as well as various intracellular organelles including mitochondria, plastids, endoplasmic reticulum, and secretory vesicles. Liposomes are ideal candidates for serving as bio-compatible micro-environments, since one can encapsulate biomolecules in their interior and membrane proteins (e.g., membrane pores) within the bilayer. A temperature-regulated coacervation process was recently demonstrated inside large liposomes (~100 μm diameter), which were produced using glass capillary devices[17]. However, this system did not take advantage of the functional benefits of membranous structures, such as their selective permeability due to active or passive protein pores. We devise a possible route to provide exquisite control over phase separation/coacervate formation inside liposomes. One feasible approach is to encapsulate one of the essential coacervate components ($C_1$) inside the liposome and embed protein nanopores in the lipid bilayer. This will allow the size-selective diffusion-mediated transport of remaining components ($C_2$) into the liposomal lumen, thereby providing external control over the onset and process of coacervate formation.

In the present paper, we show a high-throughput microfluidic on-chip methodology to form untethered, micron-sized (~2 μm in diameter) complex coacervates within cell-sized (~10 μm in diameter) lipid vesicles (Fig. 1a). It allows the production and storage of thousands of liposomes, inside which coacervation can be induced simultaneously to form hybrid microcontainers. We form

the liposomes using recently developed on-chip microfluidic method, Octanol-assisted Liposome Assembly (OLA)[18]. We encapsulate one of the coacervate components inside the liposomes and embed α-hemolysin pores in the bilayer. Addition of further components in the external environment allows passive transport into the lumen, thereby commencing the process of coacervate formation. We demonstrate the viability of the strategy to form coacervates-in-liposomes using two different systems: (i) Encapsulating a polycationic polyelectrolyte, such as the protein polypeptide (poly-L-lysine, pLL), and allowing the diffusive transport of multivalent nucleotides (adenosine triphosphate, ATP), which serve as the polyanionic component for complex coacervation. (ii) Encapsulating an enzymatic reaction that forms a polyanionic polymer (polyU RNA) and allowing diffusive transport of both the substrate (uridine diphosphate, UDP) and polycationic component (spermine), to form polyU/spermine coacervates. In both these cases, freely diffusing coacervates are formed inside the vesicles, where they remain stable for long times (>hours). Analysis of the condensation dynamics shows a spontaneous homogeneous nucleation in case of pLL/ATP and a heterogenous one in case of polyU/spermine. We furthermore show the potential functionality of the coacervates as rudimentary membraneless organelles by sequestering protein molecules (FtsZ, a key protein for bacterial cell division) as well as supramolecular assemblies such as small unilamellar vesicles (SUVs, a lipid source to potentially achieve liposome growth and form membranous sub-compartments). We also demonstrate that it is possible to conduct specific enzymatic reactions inside these synthetic organelles by showing the β-galactosidase-catalyzed degradation of a non-fluorescent substrate into a fluorescent product. Summing up, our liposome-based platform provides spatiotemporal control to study the process of forming functional coacervates in sub-picoliter confinements.

The described methodology to build hybrid systems comprised of membranous and non-membranous scaffolds also opens further avenues to increase the complexity in bottom-up synthetic biology, where one of the major goals is to establish a synthetic cell from molecular components. Hybrid coacervate-in-liposome systems can potentially be endowed with unique properties that arise due to the complementarity of their constituents[19–21]. While liposomes can effectively encapsulate molecules and form transmembrane gradients, coacervates enable a local heterogenous increase of charged and hydrophilic molecules[22]. Thus, their combination may allow for the creation of synthetic cells with a degree of heterogeneity that typically is observed in living cells.

## Results

**On-chip experimental set-up to study coacervate dynamics**. We set out to induce the controlled formation of membraneless coacervates inside liposomes (Fig. 1a). The idea was to encapsulate part of the necessary components ($C_1$) inside the liposomes and allow the transport of the remaining necessary component ($C_2$) through protein pores in the membrane. We produced cell-sized (10–15 μm in diameter) unilamellar liposomes using OLA[18]. We separated the formed liposomes from the waste product (less dense 1-octanol droplets) using a modified version of a density-based separation technique that we have previously reported[23,24]. We implemented two major changes (Fig. 1b): Firstly, we punched a large collection well at the end of the production channel in order to collect liposomes at its bottom and let the waste product (1-octanol) float to the top of the buffer-filled well. Secondly, we made the liposomes slightly denser than the environment by encapsulating disaccharides (sucrose) or polysaccharides (dextran), to induce sedimentation of the liposomes within a few minutes. 50–100 mM sucrose or 3–5 mM dextran (molecular weight (MW) 6000) was observed to be optimal in

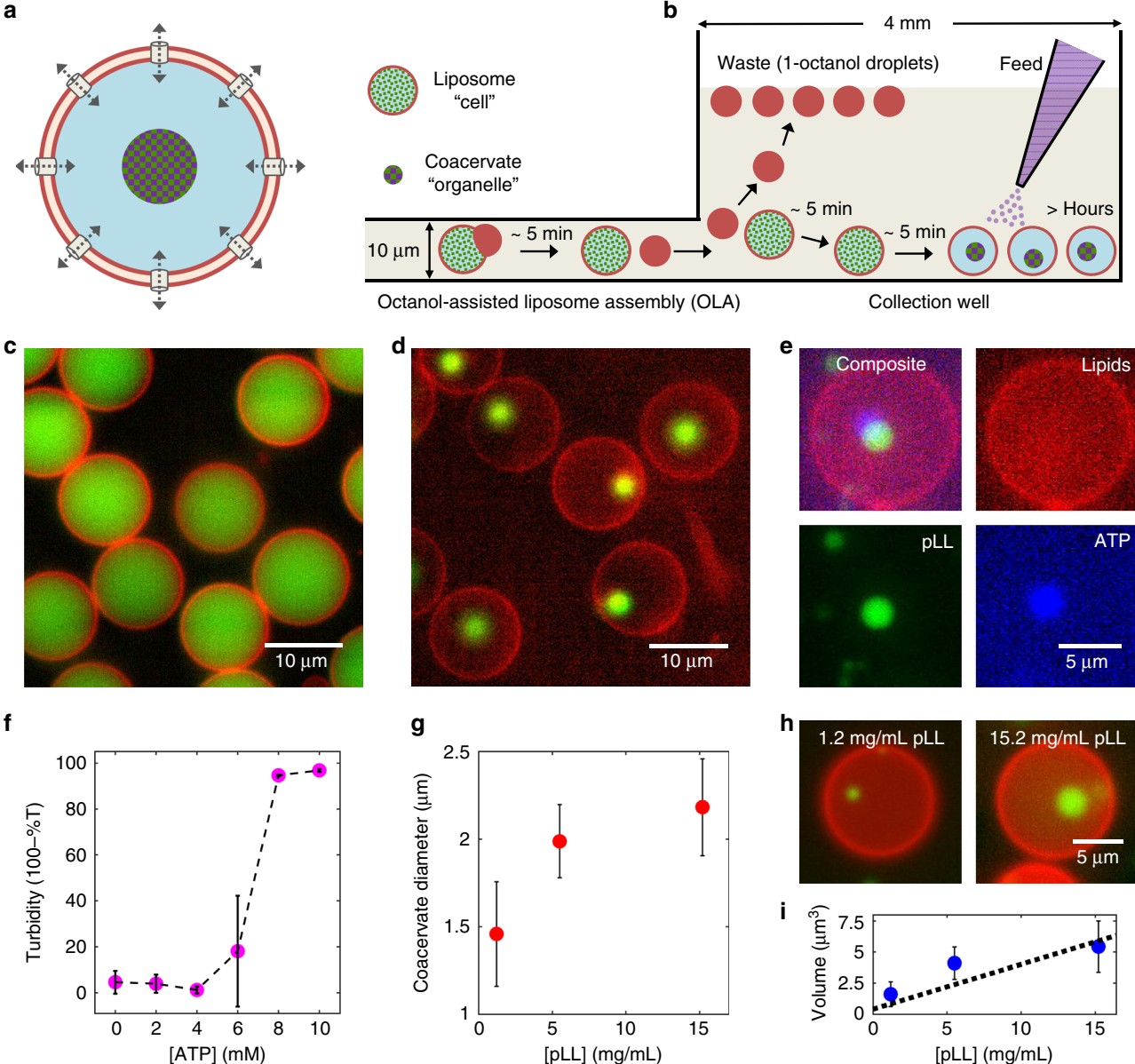

**Fig. 1** Controlled formation of membraneless coacervates in liposomes. **a** A conceptual sketch showing controlled condensation within a liposome. By inserting bilayer-spanning protein pores, one can allow passive transport of small molecules, leading to the formation of a coacervate. Such a coacervate-in-liposome hybrid system can also be used as a scaffold for a synthetic cell, where the liposome represents the primary compartment (a cell), while the coacervate represents a sub-compartment (an organelle). **b** Side-view schematic (not to scale) of the experimental set-up. Liposomes encapsulating one of the coacervate component ($C_1$) or an enzyme catalyzing the production of $C_1$ are generated using OLA. The presence of high-density molecules, such as dextran, efficiently settles them at the bottom of the collection well, while the waste products (1-octanol droplets) float to the top. The other coacervate component, $C_2$, can already be present in the chamber or can be added later to induce coacervation. **c** A fluorescence image showing monodisperse liposomes settled at the bottom of the well. The red boundary indicates the lipid bilayer while the green lumen shows encapsulated FITC-pLL molecules. **d** A fluorescence image showing coacervates-in-liposomes. Transport of ATP through α-hemolysin pores led to a single pLL/ATP coacervate within each vesicle. **e** A composite image showing the colocalization of FITC-pLL (green) and cy5-ATP (blue), forming a coacervate within the liposome (red). The slight offset between pLL and ATP fluorescence is due to the diffusion of the coacervate between capturing of the images. **f** Turbidity plot showing that the threshold ATP concentration to commence coacervation is about 6 mM. Absorbance of three independent samples was measured (nine measurements per sample). **g** Dependence of coacervate size on the amount of pLL molecules encapsulated inside the liposome ($n \geq 73$ for each data point). **h** Representative images of coacervates of different sizes formed within liposomes with different pLL concentrations. **i** Dependence of coacervate volume on pLL concentration. The coacervate volume is seen to scale approximately linearly ($R^2 = 0.78$) with the pLL concentration. Error bars in (**f**, **g**, **i**) indicate standard deviations. Source data are provided as a Source Data file

settling the liposomes quickly and effectively at the bottom of the well (Fig. 1c). Dextran was the better choice, because of the low concentration that was sufficient and because of its inability to diffuse out of the membrane pores due to its large size. This improvement of encapsulating dense molecules to induce the settling of liposomes, led to their complete isolation from unwanted side-products of OLA and allowed straightforward long-term experimentation. With this experimental set-up, we investigated the formation of hybrid coacervate-in-liposome systems.

**Coacervation within liposomes via transport through pores.** To study coacervate formation, we started with a protein polymer/nucleotide system, where we encapsulated positively charged pLL polymers (4.5–5 mg/mL pLL, MW 15–30 kDa) inside the liposomes, and added negatively charged ATP molecules (25 mM) in the surrounding environment. α-Hemolysin proteins were added to the outside (or alternatively encapsulated inside) of the liposomes, which spontaneously insert into the membrane to form ~1.4 nm diameter pores that allow diffusive transport of small molecules (<2 kDa), such as ATP, into the vesicles[25,26]. By using fluorescently labeled pLL molecules (0.5 mg/mL FITC-pLL, MW 15–30 kDa or 0.25 mg/mL cy5-pLL, MW ~25 kDa), we observed the formation of coacervates inside the liposomes (Fig. 1d). The formation was induced within a few minutes after the liposomes were exposed to the ATP-containing environment. The probability of forming a hybrid container was $0.76 \pm 0.18$ (mean ± standard deviation, data obtained from six independent experiments; see Methods for details), suggesting efficient hybrid container formation. In the absence of protein pores in the membrane, no coacervation was observed in majority of the liposomes, even after 30 min (Supplementary Fig. 1). A minor fraction (11%, $n_{total} = 539$) did develop coacervates, which we attribute to non-specific membrane defects and the possible formation-resealing of transient pores[27,28] due to the shear experienced when liposomes entered the collection well. We also noted that most not-yet-fully-matured liposomes, i.e., those with protruding 1-octanol pockets, formed coacervates (96%, $n_{total} = 204$), even in the absence of protein pores (Supplementary Fig. 1). This effect suggests a possible transport of ATP across the contact region between the liposome and the 1-octanol pocket, as 1-octanol is known to affect membrane tension as well as membrane permeability[29–31].

To confirm the intake of ATP from the environment leading to the observed LLPS, we used fluorescently labeled ATP (2.5 μM cy5-ATP) to induce coacervation. Colocalization of the ATP signal with that of the pLL in the condensed phase clearly proved that the diffusion of ATP through the membrane pores led to the formation of coacervates (Fig. 1e). We also performed a bulk turbidity assay to estimate the ATP concentration above which coacervation took place (see Methods). Under the given buffer conditions, the threshold ATP concentration was observed to be about 6 mM (Fig. 1f; three independent samples, with nine measurements per sample). Consequently, as soon as the diffusive transport across the porous membrane increased the internal ATP concentration above this threshold, coacervation took place.

The microfluidic set-up allowed us to obtain monodisperse coacervate-containing liposome samples. For example, within a single experiment, we measured on a monodisperse population of liposomes ($d_{vesicle} = 14.2 \pm 0.9$ μm, $n = 213$) and obtained monodisperse coacervates that were formed within them ($d_{coacervates} = 2.8 \pm 0.3$ μm, $n = 213$). Indeed, the corresponding coefficients of variation of respectively 6% and 11% indicate a low degree of variability. Since there is virtually unlimited supply of ATP (as the volume of the well, ~30 μL ≫ volume of a liposome, ~0.5 pL), one would expect that the size of the formed coacervates is set by the finite amount of pLL molecules present inside the liposomes. We tested this by varying the pLL concentration inside the liposomes and measuring the size of the formed coacervates (see Methods for details). As can be seen in Fig. 1g, the mean coacervate diameter indeed increased with the amount of encapsulated pLL; cf. representative images shown in Fig. 1h. As expected, we obtained an approximately linear relationship ($R^2 = 0.78$) between the coacervate volume and the pLL concentration (Fig. 1i). We also checked whether dextran, used to settle the liposomes, accumulated in the coacervates. We tested this using fluorescently labeled dextran (AF647-dextran). The

results showed no significant sequestration of dextran molecules inside the coacervates but instead showed a similar fluorescence intensity as in the rest of the liposomal lumen, with a slight accumulation at the coacervate interface (Supplementary Fig. 2). This observed accumulation at the interface clearly did not alter the coacervation process, as the coacervation dynamics remained unchanged, independent of whether we used sucrose or dextran to settle the liposomes. We also note that technical problems in liposome production, such as an undesirable bursting of double-emulsion droplets, can release some encapsulated coacervate components (e.g., pLL) into the collection well, leading to a residual amount of unwanted coacervation outside the liposomes, as can for example be noted in Fig. 1e. A detailed troubleshooting to ensure stable liposome production can be found in our online protocol[24].

Our approach enables to measure the time dependence of the coacervation process. We captured time-lapse movies by gently introducing the ATP-containing solution into the well after the liposomes were settled to the bottom. The process is shown schematically in Fig. 2a, and the time-lapse images, visualized by the fluorescence of FITC-pLL, are shown in Fig. 2b (also see Supplementary Movie 1). ATP molecules rapidly diffused throughout the well, and entered the liposomes through the pores. Once the threshold ATP concentration required for coacervation was reached inside the vesicle, a rapid phase transition was observed. Small, discrete condensates appeared throughout the liposomal lumen, with a bright fluorescence indicating a high concentration of pLL molecules inside them. These small condensates fused with each other—incidentally confirming their liquid nature—to form a single coacervate (Fig. 2c). Importantly, this single liquid droplet continued to diffuse inside the liposome without adhering to the inner leaflet of the lipid bilayer. The coacervation process did not commence simultaneously in the entire liposome population, but its onset was randomly distributed and occurred over a period of about 10 min. We observed coacervates within 88% ($n_{total} = 218$) of the liposome population.

In order to quantify the coacervation process, we analyzed the number and the average size of the coacervates formed as a function of time as well as the time evolution of the fluorescence intensity belonging to the condensed phase and that belonging to the dilute phase (Fig. 2d–g, see Methods for details). The start of the coacervation process was marked by the appearance of multiple (~8) small coacervates, throughout the liposomal lumen (Fig. 2d, $n = 12$). After this homogenous nucleation event, the formed small coacervates rapidly fused with each other, reducing the number of freely diffusing coacervates to ~4 over the next 2 min. The average number further halved within the next 3 min and already resulted in a single, large coacervate in a few cases. Complementary to the decrease in the number, the average coacervate size increased steadily from 0.8 μm to about 2.1 μm (Fig. 2e). The start of the coacervation was also clearly observed in the form of a sudden rise in the coacervate-phase intensity, a quantity corresponding to the amount of phase-separated coacervate material based on fluorescence intensity (Fig. 2f). The value plateaued within 2 min and stayed constant thereafter, thus not getting affected by the latter coalescence of coacervates into a single entity. Next, we attempted to detect the nucleation events that led to the formation of condensates by plotting the dilute-phase intensity over time, a quantity which corresponds to the less-bright fluorescence background intensity inside the liposomes. We observed a rapid decay, complementary with the formation of the coacervate phase (Fig. 2g). This is expected, as coacervation considerably decreases the fraction of pLL molecules that reside in the dilute phase, decreasing the background intensity in the liposomal lumen. More interestingly, we identified

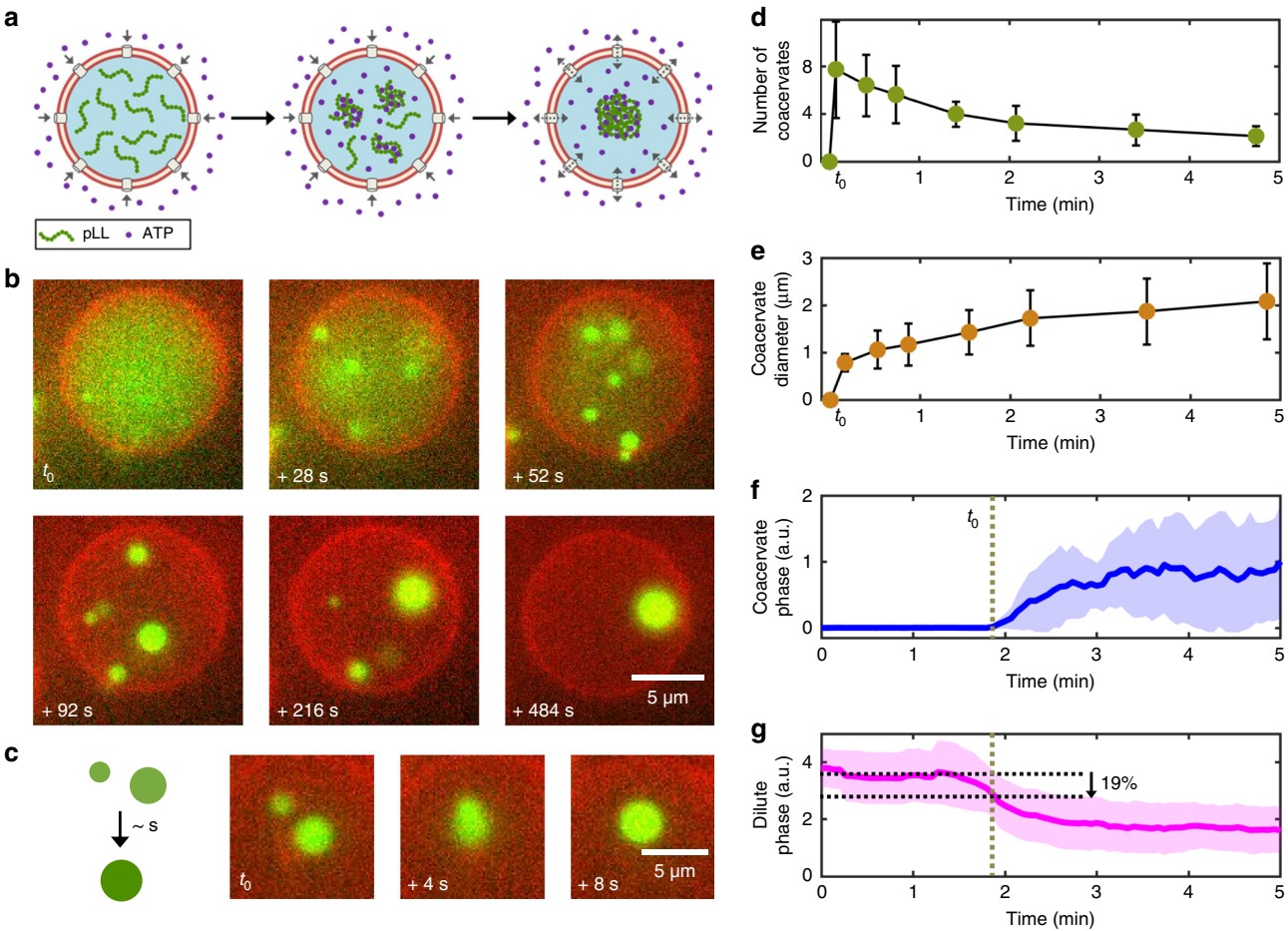

**Fig. 2** pLL/ATP coacervate formation within liposomes through influx of ATP. **a** A schematic showing the formation of a pLL/ATP coacervate within a liposome. ATP from the environment diffuses inside the liposome through α-hemolysin pores. It interacts with pLL molecules present inside the liposome, initiating coacervation throughout the liposome. Over time, individual coacervates coalesce to form a single coacervate. **b** Time-lapse fluorescence images (red: lipid bilayer; green: FITC-pLL) showing the coacervation process inside a liposome. Note the simultaneous formation of multiple coacervates that further coalesce to form one single entity. Time zero ($t_0$) corresponds to the first visible sign of coacervation. **c** Schematic and a typical example showing coalescence of two coacervates, a characteristic feature of liquid droplets. **d** Average number of coacervates present inside a liposome as a function of time. After the initial burst of multiple coacervates (~8) throughout the liposome, the average number decreased to about 2 within the next 5 min. **e** Evolution of the average coacervate size with time. Initially formed small coacervates (~1 μm) subsequently fuse with each other, to ultimately form a ~2 μm coacervate that freely diffuses inside the liposome. **f** Coacervate-phase intensity versus time ($n = 12$ liposomes, obtained from a single experiment). The rapid transition from a homogeneous solution to a condensed phase leads to sudden rise in the fluorescence intensity, which plateaus over time. **g** A plot of the dilute-phase intensity over time ($n = 12$), showing a complementary rapid decay. The horizontal dashed lines indicate the decrease in the fluorescence intensity just before coacervation takes place, providing evidence for nucleation events. See Methods for details of the analyses involved for panels (**d**–**g**). Error bars in (**d**, **e**) indicate standard deviations. Dashed vertical lines in (**f**, **g**) indicate the onset of coacervation, the plots show the average values, with the shaded regions indicating standard deviations. Source data are provided as a Source Data file

a short regime (~30 s), prior to the emergence of first coacervates (before $t_0$), that showed a marked intensity decrease of the dilute phase. We speculate that this fluorescence intensity loss, just before the coacervates are observed, indicates the nucleation process that precedes the observable formation of coacervates. The ~20% decrease at the moment of appearance of visible structures would mean that about 20% of the pLL molecules had already nucleated before any visibly detectable coacervates. The observed changes in the fluorescence intensity were not a result of photobleaching, as the total fluorescence counts of a liposome not showing coacervation stayed constant over a similar period of time (Supplementary Fig. 3).

Along with capturing the coacervation dynamics, we simultaneously assessed the continuous interaction of the formed coacervate with the surrounding dilute phase. We did this by encapsulating apyrase inside the liposome, an enzyme that

degrades ATP into adenosine diphosphate (ADP) and finally into adenosine monophosphate (AMP). ADP or AMP cannot coacervate with pLL in the presence of a sufficient concentration of screening ions (150 mM KCl and 5 mM MgCl₂ in our case, see Supplementary Fig. 4). One would thus expect the coacervate to dissolve over time in the presence of apyrase, unless the degraded ATP was constantly replenished from the environment. Indeed, a bulk experiment, where an apyrase-containing solution (without any ATP) was flown over pre-formed pLL/ATP coacervates, led to coacervate dissolution in minutes (Supplementary Movie 2). In contrast, the coacervates inside the liposomes remained highly stable in the presence of apyrase, even over a course of hours (Supplementary Movie 3). This showed that there was a constant exchange of coacervate material with the environment, i.e., ATP was continuously replenished. Note that in the absence of apyrase, the coacervation progressed in an exactly similar fashion,

both in terms of the dynamics and the time scale (Supplementary Fig. 5). Clearly, the degradation activity of apyrase was not strong enough to counterbalance the influx of ATP through the pores and thus did not affect the induction of coacervation. As a side note, the onset and subsequent progression of coacervation seen in a minor fraction of leaky liposomes, in the absence of membrane pores, was similar to that seen in pore-containing liposomes (Supplementary Fig. 6). Overall, we conclude that pore-permeated liposomes present a viable strategy to form stable coacervates in a controlled manner.

**Enzyme-catalyzed coacervation inside porous liposomes.** Next, we constructed a system where the influx of substrate triggered an enzymatic reaction, which subsequently induced complex coacervation (Fig. 3a). We chose a biochemical reaction catalyzed by

polynucleotide phosphorylase (PNPase), which polymerizes RNA by the incorporation of uridine monophosphate (UMP) from UDP at the 3′-end of an oligomeric RNA template[32,33]. We encapsulated PNPase along with 5′-fluorescently labeled 20-nucleotide long RNA seed oligomers (cy5-U20). UDP was provided externally, and spermine was present inside and outside of the vesicle, serving as a positively charged polyelectrolyte. UDP readily entered the liposome through the pores, and was consumed by the PNPase to form long (200–10,000 nucleotides[32]) RNA polymers. In contrast to the short cy5-U20 RNA, the elongated polyU polymers phase-separate with spermine into polyU/spermine coacervates[34] (Fig. 3b). Again, several small condensates formed simultaneously which ultimately coalesced into a single droplet (Supplementary Movie 4). We observed coacervates within 86% ($n_{total} = 159$) of the liposome population,

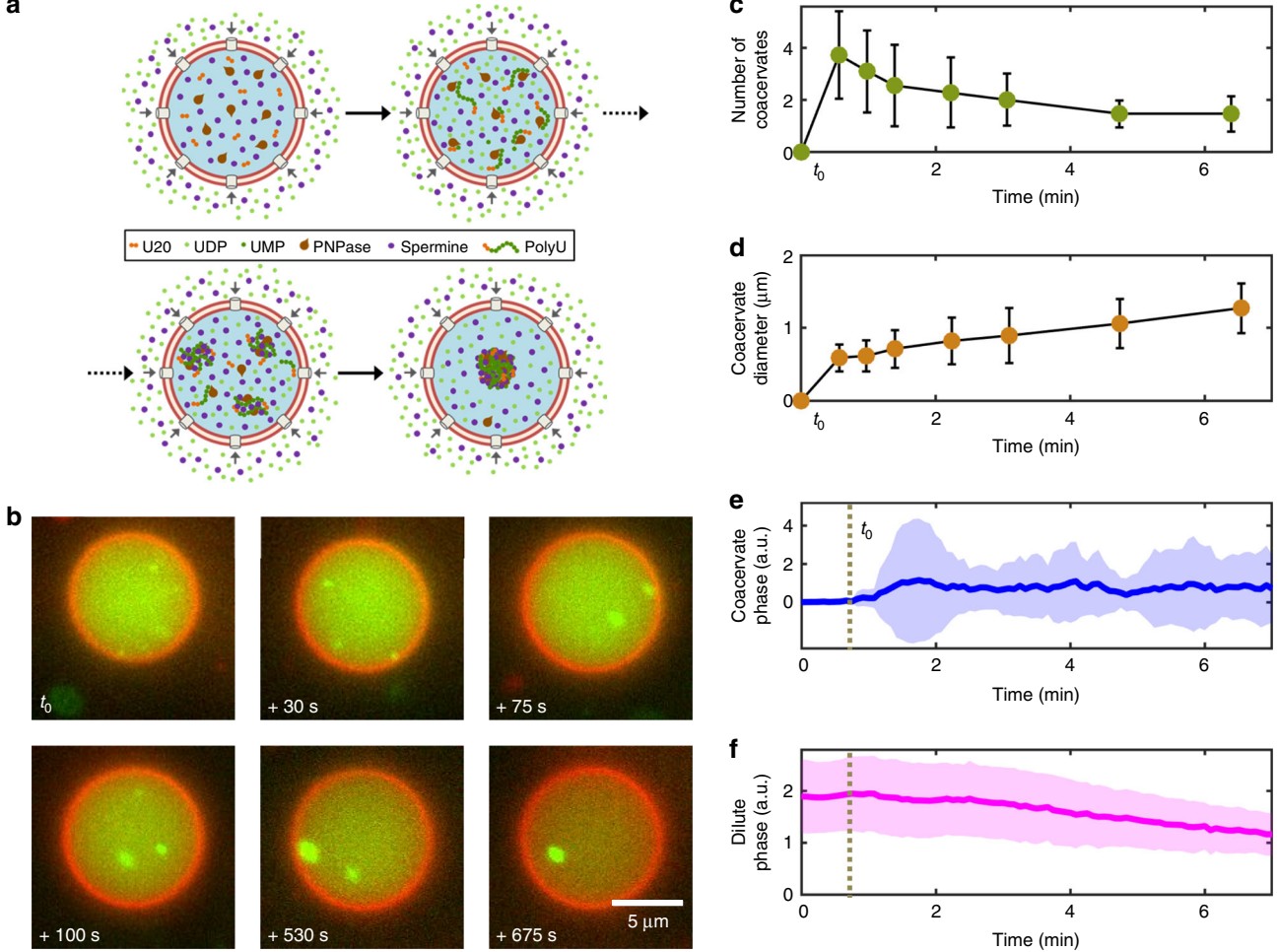

**Fig. 3** Enzymatically catalyzed polyU/spermine coacervate formation inside liposomes. **a** A schematic showing polyU/spermine coacervation within a liposome. UDP diffuses inside the liposome through α-hemolysin pores and is consumed by the enzyme PNPase to elongate encapsulated cy5-U20 seed oligomers. The formed polyU RNA further interacts with spermine to form polyU/spermine coacervates, which fuse together to form a single condensate over time. **b** Time-lapse fluorescence images (red: lipid bilayer; green: cy5-U20) showing polyU/spermine coacervation process inside a liposome. Note the simultaneous formation of several coacervates that further coalesce to form one single entity. Compared to pLL/ATP coacervation, the process proceeds slowly, as it is primarily dictated by the RNA elongation process. **c** Average number of coacervates present within the liposomes as a function of time. A few coacervates (~4) are observed at the start, whereupon the average number decreases to about 1.3 within the next 5 min. **d** Evolution of the average coacervate size with time. Initially formed small coacervates (~0.6 μm) more than double in their diameter through growth and fusion, to ultimately form a ~1.5 μm coacervate that is freely diffusing inside the liposome. **e** Coacervate-phase intensity versus time ($n = 11$ liposomes, obtained from a single experiment). The transition from a homogeneous solution into a condensed coacervate phase leads to a gradual increase in the fluorescence. **f** Dilute-phase intensity versus time ($n = 11$). As the coacervates continue to grow, there is a gradual, linear decrease in the fluorescence of the dilute phase. See Methods for details of the analyses involved for panels (**c**–**f**). Error bars in (**c**, **d**) indicate standard deviations. Dashed vertical lines in (**e**, **f**) indicate the onset of coacervation, the plots show the average values, with the shaded regions indicating standard deviations. Source data are provided as a Source Data file

suggesting the same level of efficiency as observed for the pLL/ATP system. Note that the polyU/spermine system is fundamentally different from the pLL/ATP system: in the case of pLL/ATP coacervates, both components were already present at a constant concentration and the reaction was mainly limited by the diffusion of ATP into the container. In case of polyU/spermine, however, one of the components (polyU) was produced locally, which simultaneously triggered the coacervation process. Also, the rate-limiting step was PNPase activity rather than diffusion of UDP into the coacervate, and as a result, the coacervation dynamics were, in general, slower.

This inherent difference between the two systems (RNA/spermine versus pLL/ATP) became further evident by doing similar analyses as in case of pLL/ATP system (Fig. 3c–f, see Methods for details). The number of coacervates formed initially (~4) was much lower than that observed for the pLL/ATP system (Fig. 3c, $n = 11$), suggesting heterogeneous nucleation where coacervation commenced at places with locally high polyU concentration due to the ongoing local PNPase activity. The average coacervate size increased from ~0.5 µm in the beginning to about 1.5 µm after 6 min, when the majority of the liposomes already contained a single coacervate (Fig. 3d). Thus, the size of the coacervates remained relatively small as compared to the pLL/ATP system. We observed that the coacervates remained stable and did not increase in size even after a few hours (Supplementary Movie 5, Supplementary Fig. 7). With virtually unlimited supply of spermine and UDP, one would expect the coacervates to continuously grow. However, a limited polymerization activity of PNPase and activation of the RNA exoribonuclease activity of the enzyme[33] possibly inhibited further growth. A time-series plot of the coacervate-phase intensity also showed an altogether different behavior compared to the pLL/ATP system: a gradual increase was observed for an initial time period of about 1 min, corresponding to the formation of multiple small coacervates (Fig. 3e). This lack of a burst-like nucleation pattern fits the gradual synthesis of polyU RNA polymer which starts forming a coacervate only after sufficient elongation. The fluorescence intensity remained relatively constant after the initial increase. The plot of the dilute-phase intensity remained relatively constant at the beginning and subsequently showed a monotonous decrease, without any discontinuous transition as was seen in the case of pLL/ATP coacervation (Fig. 3f). The observed changes in the fluorescence intensity did not result from photobleaching, as confirmed by the constant total fluorescence counts of a liposome not showing coacervation, over a similar period of time (Supplementary Fig. 3). In conclusion, we successfully triggered the formation of stable condensates inside liposomes, through the influx of substrate that induced a biochemical reaction and subsequent complex coacervation.

**Sequestration and compartmentalization in hybrid containers.** After successfully forming condensates inside vesicles using two different systems (pLL/ATP and polyU/spermine), we set out to demonstrate the functional utilities of such hybrid containers. We focused on two salient and well-known features of condensates that bring out their potential to act as membraneless organelles: the specific sequestration of biomolecules and their use as reaction centers[6,10,12,14,34]. We used pLL/ATP condensates (5 mg/mL pLL and 0.25 mg/mL cy5-pLL encapsulated inside, with 10 mM ATP present in the external environment) for the experiments that follow.

First, we encapsulated FtsZ protein inside the liposomes. FtsZ is a bacterial protein crucially involved in the process of cell division[35]. We observed that FtsZ sequestration occurred in a highly efficient way. Its concentration was concomitant with the coacervate formation and the FtsZ fluorescence was distributed homogenously throughout the coacervate (Fig. 4a). As a means to quantify the degree of protein localization in a host condensate, we measured the partition coefficient of the protein to be $P_{FtsZ} = 14.2 \pm 3.8$ ($n = 25$, see Methods for details). The observed homogenous distribution contrasts the FtsZ localization at the interface of pLL/ATP coacervates that was reported recently[11], where FtsZ was added to a solution containing pre-formed, stable coacervates. In the current case, however, FtsZ was already present when the phase separation occurred. We observed a similar homogenous distribution of FtsZ inside the coacervates, when the coacervates were prepared in the absence of any vesicles (Supplementary Fig. 8). This ruled out any effect of the membranous confinement on the spatial organization of FtsZ within the coacervate. These experiments suggest that the spatial organization of sequestered molecules within the coacervate depends on the sequence and timing of the addition of parts.

Next, we probed the possible sequestration of SUVs (~30 nm diameter), self-assembled supramolecular lipid assemblies. Encapsulating the SUVs inside liposomes and triggering pLL/ATP coacervation did concentrate the SUVs within the coacervates (Fig. 4b). We measured a partition coefficient $P_{SUVs} = 2.9 \pm 0.6$ ($n = 35$), indicating that the coacervates acted as an efficient lipid reservoir. The SUVs were distributed homogenously within the coacervate, as opposed to previously reported localization of SUVs at the interface of polyU/spermine coacervates, where the coacervates were prepared before adding the SUVs[34]. A similar homogenous SUV distribution was obtained, when the coacervates were prepared in the absence of any vesicles, ruling out any effect by the compartmentalization (Supplementary Fig. 9). Our experiments with FtsZ and SUVs suggest that the sequence of reactions plays an important role in determining the nature of sequestration. While sequential addition of components favors the accumulation of material at the coacervate–solvent interface, co-condensation is favored when the biomolecules are present before coacervation is triggered through external cues.

Finally, we investigated the possibility to carry out a biochemical reaction specifically inside the coacervates. We chose the enzyme β-galactosidase, which converts β-galactosides into monosaccharides by cleaving the glycosidic bond. To monitor the enzyme activity, we used fluorescein di-β-D-galactopyranoside (FDG), a non-fluorescent substrate that upon cleavage by β-galactosidase releases fluorescein and thus induces fluorescence. We encapsulated β-galactosidase, along with pLL, inside liposomes, and induced coacervate formation by allowing the ATP molecules from the external environment to diffuse in through the protein pores. In the absence of FDG in the environment, we did not detect any enzymatic activity, based on the lack of fluorescent signal (Fig. 4c). However, the presence of FDG molecules in the external solution resulted in their diffusion through the porous membrane to trigger the enzymatic reaction, as seen by the increase in the fluorescence emitted by fluorescein (Fig. 4c). While we observed fluorescent signal throughout the liposome, the signal obtained from the coacervate phase was clearly stronger than the one obtained from the surrounding dilute phase, which indicates that β-galactosidase had a higher affinity to reside within the coacervate phase. The partition coefficient of fluorescein after 2 h of reaction time was found to be $P_{fluorescein} = 1.3 \pm 0.1$ ($n = 31$). The fluorescein intensity within the coacervate phase increased roughly 14-fold over a course of 2 h (Fig. 4d, $n = 31$ in both the cases). The reaction product (fluorescein) likely remained confined within the condensed phase due to hydrophobic stabilization and potential electrostatic interactions with cationic groups of pLL[12]. Performing the experiment in bulk, in the absence of any confinement, led to a

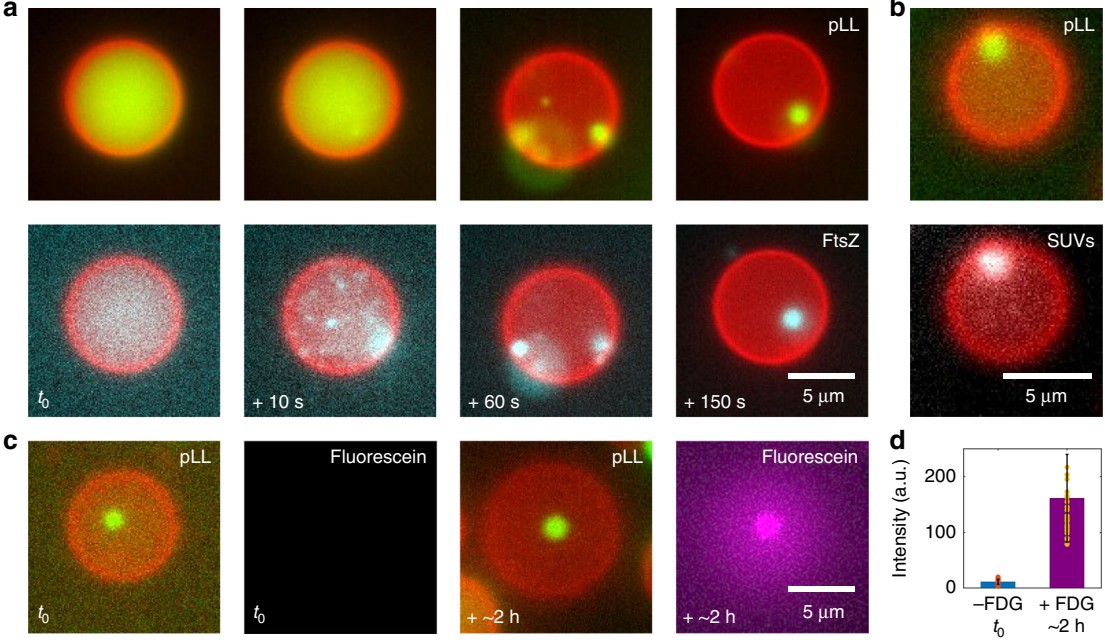

**Fig. 4** Sequestration of biomolecules and compartmentalization of reactions within coacervates. The pLL/ATP coacervate (green) formed within the liposome (red) serves multiple purposes, from concentrating proteins and supramolecular assemblies to facilitating enzymatic reactions. **a** Fluorescence time-lapse images showing coacervation dynamics (upper panel) and simultaneous sequestration of FtsZ (cyan) within the coacervates (lower panel). **b** SUVs (in white) get sequestered inside the condensates. **c** The β-galactosidase reaction is getting carried out predominantly within the coacervate. No fluorescent product is observed before the addition of the FDG substrate ($t_0$)—two left panels. Addition of FDG induces the enzymatic reaction inside the liposome, particularly in the condensed phase, as can be seen from the colocalization of the fluorescent product, fluorescein (in magenta) with the coacervate—two right panels. **d** Fluorescein intensity within the coacervate phase increases about 14-fold in 2 h after the addition of the substrate ($n = 31$ in both the cases). The circles represent the individual data points, bars represent the mean fluorescence intensity, and the error bars indicate corresponding standard deviations. Source data are provided as a Source Data file

similar result (Supplementary Fig. 10): We observed an increase in the fluorescence intensity in both the dilute and the condensed phase, but always with a higher intensity in the latter, which again suggested that β-galactosidase was partitioned more into the condensed phase. Summing up, we successfully demonstrated two important attributes of coacervates: sequestering and concentrating biomolecules, and serving as hubs for biochemical reactions.

## Discussion

In this paper, we reported the controlled formation of hybrid microcontainers, viz., coacervates-in-liposomes. We show that these are ideal systems to study the dynamics of LLPS, and are also well suited as potential architectural scaffolds for the design of future synthetic cells. We used two different (poly)nucleotide–(poly)peptide/(poly)amine systems to form freely diffusing coacervates, a model system for membraneless organelles within the liposomes. The nucleation of condensates inside sub-picoliter liposomes was triggered by the diffusive transport of small polyelectrolytes or coacervate precursors into the liposomes through α-hemolysin protein pores embedded in the liposomal membrane.

We employed the on-chip microfluidic method OLA[18] to produce liposomes, where a collection well at the end of the production channel allowed to immediately settle and visualize the liposomes, a process facilitated by making the liposomes denser by encapsulating sucrose or dextran. With this platform at hand, we obtained temporal control over the onset of coacervation as well as the final size of condensates. The temporal control was obtained by the addition of essential components for condensation from the outside, and their entry into the liposomes through protein nanopores present in the lipid bilayer, in order to trigger LLPS. Control over the condensate size was achieved by

encapsulating a precise and finite amount of material within the liposome. The ability to induce and selectively limit the coacervation process presents a crucial advantage for experiments in which spatiotemporal control and monitoring is desirable. For example, using fluorescence intensity analyses, we were able to detect the de novo nucleation of condensates in real time, a process which has been difficult to study and control so far[1,36]. We were further able to follow and compare the dynamics of two different coacervate systems to demonstrate the strength of our experimental setting: (i) A straightforward coacervation reaction mediated through diffusive transport of a necessary component across the membrane. (ii) A more complex scenario where real-time production of a multivalent polymer subsequently triggered the coacervation process. We analyzed and quantified the differences between the two systems, in terms of the evolution of the condensed and the dilute phase, as well as the number and size of the formed coacervates. Our on-chip set-up is ideally suited to follow coacervation dynamics (formation, dissolution, reentrant transitions) in a micro-confinement, without any unwanted surface interactions or other external interfering agents. The confinement volume can be changed by modulating the liposome diameter, and surface interactions, if desired, can be introduced using charged lipids. The approach also allows to modulate a variety of reaction parameters (pH, salt concentration, etc.), and study a variety of phenomena relevant to intracellular phase transitions such as concentration buffering, signal amplification, and spatiotemporal regulation[7].

We used a well-known polypeptide-nucleotide system (pLL/ATP) to demonstrate the formation of coacervates inside liposomes. We observed that the condensation process could be divided into two distinct regimes: (i) Spontaneous and sudden nucleation transition in which condensation was triggered at

several locations throughout the liposome, giving rise to multiple coacervates. The nucleation was primarily dictated by the influx of ATP through the membrane pores. After a threshold concentration of ATP (~6 mM) was reached, spontaneous LLPS was observed throughout the liposomal lumen. (ii) Coalescence regime, in which a single, large condensate was formed as a result of the fusion of small coacervates. The signature of the second regime was the diffusive motion of coacervates and their coalescence, consisting of consecutive merging events towards one single coacervate. We further confirmed the stability of the formed coacervates by promoting an enzymatic ATP degradation reaction within the liposome. The highly stable nature of the coacervates indicated that there is a continuous exchange of molecules between the coacervate and their surroundings, as expected for liquid–liquid phase separated mixtures.

Thereafter, we demonstrated a more complex enzyme-catalyzed coacervation process. By encapsulating PNPase, an enzyme that efficiently produces long (>200 nucleotides) single-stranded RNA molecules from short primers and nucleotide diphosphates, we induced polyU/spermine coacervation. Compared to the pLL/ATP coacervation, the condensation process was more gradual in the RNA/spermine mixture, as it was limited by the RNA polymerization rate. The coacervates did not grow indefinitely but soon reached a stable size, despite the fact that neither of the coacervate components were limited. This could be a result of a limited polymerization activity of PNPase along with the activation of the exoribonuclease activity of the enzyme[33], which would become more pronounced over time as the polyU concentration increased, degrading the long polyU polymers into progressively shorter fragments and making them unsuitable for coacervation. Thus, we found pronounced differences between the pLL/ATP and RNA/spermine condensation dynamics, suggesting that they differ fundamentally in their nucleation behavior and dynamics. Since both the components were already present in case of pLL/ATP system, the nucleation was homogenous without any particular spatial preference. In case of polyU/spermine system, however, coacervation could only commence at a position where PNPase had synthesized enough polyU to trigger LLPS. The pLL/ATP and PNPase-induced polyU/spermine coacervation may thus be regarded as cases of homogeneous and heterogeneous nucleation, respectively[37].

Next to its importance as a tool to examine the process of coacervation with excellent spatiotemporal control, our study also has potential to impact research directed at creating synthetic cells. Architectural scaffolds of sufficient complexity and versatility are essential for designing artificial cells to exhibit the basic characteristics and dynamic organization of natural cells. We showed two main functional utilities of coacervate-in-liposome scaffolds: sequestration/concentration of biomolecules and the compartmentalization of biochemical reactions. We achieved, for example, successful sequestration of FtsZ, an important bacterial division protein, in coacervates. Possible applications of condensates in this context may be the temporal control of supply and release of protein and necessary metabolites from the coacervate. In this way, the process of liposome constriction will commence only when FtsZ is released. Similarly, we showed sequestration of SUVs inside the coacervates. The release of SUVs at a given time point can be used to feed the membrane and control membrane composition, eventually leading to liposome growth or form membranous sub-compartments. A possible strategy for the release of materials would be to re-dissolve the coacervate. This could be achieved in many ways: enzymatically degrading one of the coacervate components[16], altering the charge density of the components[15], increasing the salt concentration[16], altering temperature-dependent component interactions[17,34], or through reentrant phase transition[38]. Lastly,

we showed a compartmentalized biochemical reaction, β-galactosidase converting its non-fluorescent substrate into fluorescein that remained confined within the coacervate. This exemplary reaction demonstrates the potential to carry out chemical reactions, even multiple ones that are mutually incompatible with each other, within the same liposome but residing in different microenvironments.

Future research may expand in a variety of directions. For example, in situ observation of nucleation processes is challenging due to their intrinsic stochastic nature[39]. The possibility to observe such processes in a microcontainer with high spatiotemporal resolution provides an ideal system for future studies on a variety of related phase-separation processes. Our technique may be adapted in a high-throughput manner to design synthetic molecules in order to gain control over condensate properties, potentially leading to therapeutic targets[6,40]. Furthermore, the described system of untethered coacervates confined within a semipermeable compartment can be employed to engineer gradients of coacervate material across the membrane that keep the coacervate constantly out of equilibrium, by setting up a degradation reaction inside and a regeneration reaction outside. This may, for example, allow the investigation of growth and division of liquid droplets, as recently described theoretically[41]. Lastly, our technique of creating a vesicle system with synthetic organelles has the required versatility and potential to be a valuable tool in the bottom-up construction of synthetic cells. Influx of substrate, the reaction pathway, and the coacervation process can be seen as rudimentary analogues of the uptake of nutrients, metabolism, and spatial organization, similar to what is seen in natural cells. Hence, we believe that the ability to trigger coacervate formation inside vesicles through the influx of components, biochemical reactions, and subsequent coacervation is a step forward in the bottom-up creation of a synthetic cell, and opens up interesting avenues for better understanding of LLPS and biomolecular condensates.

## Methods

**Materials and solution compositions**. Polyvinyl alcohol (PVA; MW 30,000–70,000, 87–90% hydrolyzed), glycerol, poloxamer 188 (P188), 1-octanol, KCl, MgCl₂, Tris–HCl, EDTA, sucrose, glucose, dextran (MW 6000), pLL hydrobromide (MW 15–30 kDa), FITC-pLL (MW 15–30 kDa), ATP disodium salt, ADP sodium salt, apyrase from potato, cy5-U20 (labeled on 5′-end), UDP disodium salt, spermine tetrahydrochloride, polynucleotide phosphorylase (from Synechocystis sp.), and β-galactosidase aqueous glycerol suspension (from *Escherichia coli*) were purchased from Sigma-Aldrich. Cy5-pLL (MW ~25 kDa) was bought from Nanocs Inc. N6-(6-amino)hexyl-ATP-Cy5 was bought from Jena Bioscience. AF647-dextran (MW 10,000) and fluorescein di-β-D-galactopyranoside was purchased from Thermo Fisher Scientific. Lipids, 1,2-dioleoyl-*sn*-glycero-3-phosphocholine (DOPC), 1,2-dioleoyl-*sn*-glycero-3-phosphoethanolamine (DOPE), 1,2-dioleoyl-*sn*-glycero-3-phosphoethanolamine-N-(lissamine rhodamine B sulfonyl) (Rh-PE), and 1,2-dioleoyl-*sn*-glycero-3-phosphoethanolamine-N-(carboxyfluorescein) ammonium salt (PE-CF) were purchased from Avanti Polar Lipids. FtsZ purification and labeling (conjugated to Alexa Fluor 488, with 46% labeling efficiency) were performed in the lab following the protocol described elsewhere[42]. Protein plasmids were a kind gift from Germán Rivas (Centro de Investigaciones Biológica-CSIC, Madrid). The protein aliquots (FtsZ: AF488-FtsZ = 82:18, molar ratio) were flash-frozen in liquid nitrogen and stored at −80 °C.

The inner aqueous compositions were as follows: 15% v/v glycerol, 100 mM sucrose, 150 mM KCl, 25 mM Tris–Cl (pH 7.4), 4.5 mg/mL pLL, 0.5 mg/mL FITC-pLL, 30 units/mL apyrase (Figs. 1c–e and 2, Supplementary Movie 1–2); 15.6% v/v glycerol, 100 mM sucrose, 49.4 mM spermine, 1 mM EDTA, 16.8 mM KCl, 5.4 mM MgCl₂, 102.5 mM Tris–Cl (pH 9.0), 5 µM cy5-U20, 2.2 µM PNPase, 0.6 mM HEPES, 1.5 µM α-hemolysin (Fig. 3, Supplementary Fig. 7, Supplementary Movie 4); 15% v/v glycerol, 5 mM dextran, 150 mM KCl, 5 mM MgCl₂, 25 mM Tris–Cl (pH 7.4), 5 mg/mL pLL, 0.25 mg/mL cy5-pLL, 1.5 µM α-hemolysin (Fig. 4, with additional specific components: 10 µM FtsZ (Fig. 4a); 3.5 µM SUVs (Fig. 4b); 300 units/mL β-galactosidase (Fig. 4c)); 15% v/v glycerol, 5 mM dextran, 150 mM KCl, 5 mM MgCl₂, 25 mM Tris–Cl (pH 7.4), 5 mg/mL pLL, 0.25 mg/mL cy5-pLL (Supplementary Fig. 1); 15% v/v glycerol, 5 mM dextran, 2 µM AF647-dextran, 150 mM KCl, 5 mM MgCl₂, 25 mM Tris–Cl (pH 7.4), 5 mg/mL pLL, 0.5 mg/mL FITC-pLL, 1.5 µM α-hemolysin (Supplementary Fig. 2); 15% v/v glycerol, 5 mM dextran, 150 mM KCl, 5 mM MgCl₂, 25 mM Tris–Cl (pH 7.4), 5 mg/mL pLL, 0.5 mg/mL FITC-pLL, 1.5 µM α-hemolysin (Supplementary Fig. 5);

15% v/v glycerol, 5 mM dextran, 150 mM KCl, 5 mM MgCl$_2$, 25 mM Tris–Cl (pH 7.4), 5 mg/mL pLL, 0.2 mg/mL cy5-pLL (Supplementary Fig. 6); 15.6% v/v glycerol, 5 mM dextran, 49.4 mM spermine, 1 mM EDTA, 16.8 mM KCl, 5.4 mM MgCl$_2$, 102.5 mM Tris–Cl (pH 9.0), 5 μM cy5-U20, 2.2 μM PNPase, 0.6 mM HEPES, 1.5 μM α-hemolysin (Supplementary Movie 5).

DOPC was the lipid of choice for all the experiments. A fluorescent lipid, Rh-PE, was additionally used for visualization (DOPC: Rh-PE = 99.9:0.1, molar ratio). Lipid stock solution (100 mg/mL in ethanol) was prepared as described elsewhere[24] and was dissolved in 1-octanol to a final concentration of 2 mg/mL for experimentation.

The outer aqueous solution consisted of 15% v/v glycerol, 100 mM sucrose, 150 mM KCl, 25 mM Tris–Cl (pH 7.4), 5% w/v P188 (Figs. 1c–e and 2, Supplementary Movie 1–2); 15% v/v glycerol, 49.4 mM spermine, 100 mM sucrose, 1 mM EDTA, 5 mM MgCl$_2$, 100 mM Tris–Cl (pH 9.0), 5% w/v P188 (Fig. 3, Supplementary Fig. 7, Supplementary Movie 4); 15% v/v glycerol, 150 mM KCl, 5 mM MgCl$_2$, 25 mM Tris–Cl (pH 7.4), 5% w/v P188 (Fig. 4, Supplementary Figs. 1, 2, 5, 6); 15% v/v glycerol, 49.4 mM spermine, 22.5 mM KCl, 5 mM MgCl$_2$, 100 mM Tris–Cl (pH 9.0), 5% w/v P188 (Supplementary Movie 5).

The exit solution, i.e., the buffer dispensed in the collection well consisted of 15% v/v glycerol, 150 mM KCl, 25 mM Tris–Cl (pH 7.4) (Figs. 1c–e and 2, Supplementary Movie 1–2; with subsequent addition of MgCl$_2$ (5 mM final concentration), ATP (25 mM final concentration), cy5-ATP (2.5 μM final concentration) and α-hemolysin (3.75 μM final concentration); 15% v/v glycerol, 49.4 mM spermine, 5 mM MgCl$_2$, 100 mM Tris–Cl (pH 9.0), 1 mM EDTA (Fig. 3, Supplementary Fig. 7, Supplementary Movie 4 with subsequent addition of UDP to a 40 mM final concentration); 15% v/v glycerol, 150 mM KCl, 5 mM MgCl$_2$, 25 mM Tris–Cl (pH 7.4), 10 mM ATP (Fig. 4a–c, Supplementary Figs. 1, 2, 5, 6; Fig. 4c also contained 210 μM FDG); 15% v/v glycerol, 49.4 mM spermine, 5 mM MgCl$_2$, 100 mM Tris–Cl (pH 9.0), 1 mM EDTA (Supplementary Movie 5 with subsequent addition of UDP to a 15 mM final concentration). Varying amounts of glucose were additionally present in the outer aqueous and the exit solution with the intention to maintain isotonic conditions.

For the bulk experiment showing dissolution of pLL/ATP coacervates by apyrase (Supplementary Movie 3), 10 mg/mL FITC-pLL and 22.8 mM ATP were mixed together in equal volumes in a PDMS-coated microfluidic chamber to form coacervates. The chamber was flushed with 5 mM CaCl$_2$ solution to remove excess ATP present in the environment. Apyrase solution (75 units/mL in 4.3 mM CaCl$_2$ solution) was then flushed into the chamber, leading to the dissolution of coacervates.

**Liposome production using OLA.** For detailed working of OLA and trouble-shooting, please refer to our online protocol[24]. Briefly, the channel designs were fabricated in silicon using e-beam lithography, followed by a dry etching procedure, and surface silanization. The height of the patterned structures, measured using a stylus profiler DektakXT (Bruker Corporation), was 6.8 and 9.4 μm for two different masters. Polydimethylsiloxane (PDMS)-based microfluidic devices were prepared as described in our protocol, with the following change. Instead of having a small separation hole and a downstream exit hole, a collection well was punched using a biopsy punch (World Precision Instruments, inner diameter 4 mm), between 5 and 8 mm away from the production junction. This well later acted as the experimental chamber. Subsequent surface treatment of the channels downstream of the production junction using 5% w/v PVA solution was slightly modified: after the PVA solution had filled the entire post-junction channel, the collection well was filled with ~30 μL of the PVA solution and incubated for 5 min, before applying vacuum suction. PEEK tubing (Inacom) was preferred to flow the inner aqueous solution, especially in case of enzyme-containing solutions.

**Preparation of small unilamellar vesicles (SUVs).** SUVs were prepared by lipid film hydration and subsequent extrusion. Appropriate amounts of DOPC, DOPE, and PE-CF stock solutions were added to a round bottom flask (DOPC: DOPE: PE-CF = 700:300:4, molar ratio). Chloroform was evaporated with a gentle stream of nitrogen while simultaneously spreading the lipids to form a thin film. Remaining trace of chloroform was removed by keeping the flask under vacuum for at least 2 h in a desiccator. The obtained thin film was hydrated with a solution of 4 mM dextran and 15% v/v glycerol, to a final lipid concentration of 20 mg/mL. The hydration was facilitated by incubating at 37 °C while shaking for at least 30 min till the entire film was dispersed in the solution. The dispersed film was then sonicated for 30 min in an ultrasonic bath in order to break large aggregates and thus ease the subsequent extrusion step. A mini-extruder (Avanti Polar Lipids) was assembled and set on a heating block at 70 °C[43]. The lipid suspension was then sequentially passed through, first, a 100 nm and then a 30 nm polycarbonate track-etched membrane (Whatman) 21 times each. The extruded SUVs were then stored at 4 °C for use.

**Image acquisition and analyses.** An Olympus IX81 inverted microscope equipped with wide-field epifluorescence illumination was used to conduct the experiments, using ×10 (UPlanFL N, numerical aperture (NA) 0.30), ×20 (UPlanSApo, NA 0.75), and ×60 (PlanApoN, NA 1.45) objectives (Olympus). Fluorescence images were recorded using a Zyla 4.2 PLUS CMOS camera (Andor Technology) and a micromanager software (version 1.4.14)[44]. Images were processed and analyzed in FIJI (ImageJ) and MATLAB (Mathworks) using self-written scripts; the

codes will be made available upon request. If required, the liposomes were tracked using a combination of FIJI plug-in (HyperStackReg) and manual tracking.

The individual values used for calculating the probability of forming a hybrid container were 60% ($n_{total}$ = 205), 58% ($n_{total}$ = 293), 100% ($n_{total}$ = 334), 61% ($n_{total}$ = 994), 88% ($n_{total}$ = 218), and 89% ($n_{total}$ = 318).

For calculating the coacervate-phase and dilute-phase fluorescence intensities (Figs. 2d, e and 3c, d), the movies were aligned to have the same time point for the initiation of coacervation, and the average non-specific background intensity was subtracted from the entire movie. The maximum fluorescence value present at the start of the movie, before the onset of the coacervation, was selected as the threshold value to differentiate between the two phases. A cumulative fluorescence intensity above (corresponding to the coacervate phase) and below (corresponding to the dilute phase) the threshold was measured and these values were divided by the area of the liposome to obtain a measure for the amount of coacervate and dilute phase present at each time point. The number of coacervates were calculated for seven distinct time points for each of the movies and the areas of strictly in-focus coacervates were also measured. The corresponding diameters were calculated from the area as $d = 2\sqrt{A/\pi}$. Due to minor variations in the exact frame numbers chosen for the different movies, the time points were averaged (Figs. 2f, g and 3e, f); the maximum standard deviation for a time point was 0.4 min. It should be noted that since we are using wide-field microscopy, we are observing a limited fraction of the liposome volume and the obtained intensity values do not reflect the total fluorescence intensity of the entire liposome but a representative sample. Also, the obtained values of the coacervate phase as well as the dilute phase are potentially influenced by the out-of-focus objects to some extent.

For assessing the dependence of the coacervate size on pLL concentration, the areas ($A$) of in-focus liposomes and coacervates were obtained from the fluorescence images. The corresponding diameters were calculated from the area as $d = 2\sqrt{A/\pi}$. The data was obtained for three different pLL concentrations: 1.2 mg/mL pLL ($d_{vesicle}$ = 10.3 ± 1.4 μm, $d_{coacervate}$ = 1.5 ± 0.3 μm, $n$ = 73); 5.5 mg/mL pLL ($d_{vesicle}$ = 14.2 ± 0.9 μm, $d_{coacervate}$ = 2.8 ± 0.3 μm, $n$ = 213); 15.2 mg/mL pLL ($d_{vesicle}$ = 10.9 ± 1.1 μm, $d_{coacervate}$ = 2.4 ± 0.3 μm, $n$ = 213). The coacervate diameters were then re-scaled to a liposome diameter of 10 μm to obtain the following values, respectively: $d_{coacervate}$ = 1.5 ± 0.3 μm (1.2 mg/mL pLL); $d_{coacervate}$ = 2.0 ± 0.2 μm (5.5 mg/mL pLL); $d_{coacervate}$ = 2.2 ± 0.3 μm (15.2 mg/mL pLL).

Displayed fluorescence images are false-colored, subjected to background subtraction if needed, and contrast is enhanced for better visualization.

**Estimating the threshold ATP concentration for coacervation.** To estimate the ATP concentration above which pLL/ATP coacervation takes place, we performed a commonly used turbidity assay[15,34,45]. The absorbance ($\lambda$ = 500 nm) was measured using DS-11+ Spectrophotometer (DeNovix), for samples containing 15% glycerol, 5 mM dextran, 150 mM KCl, 5 mM MgCl$_2$, 25 mM Tris–HCl (pH 7.4), 5 mg/mL pLL, and a variable ATP concentration, ranging between 0 and 10 mM. Samples were incubated at room temperature for 15 min prior to measurements. Absorbance measurements were done against a blank containing 15% glycerol, 5 mM dextran, 150 mM KCl, 5 mM MgCl$_2$, 25 mM Tris–Cl (pH 7.4). The absorbance ($A$) was converted to turbidity as $100(1 - e^{-A})$.

**Calculation of partition coefficients.** Average fluorescence intensity of the entity in consideration (FtsZ, SUVs, or fluorescein) within the coacervate ($I_{coacervate}$), along with the average fluorescence intensity within the liposome ($I_{liposome}$), and average non-specific background intensity ($I_{background}$) were measured. The partition coefficient was then calculated as $P = (I_{coacervate} - I_{background})/(I_{liposome} - I_{background})$.

**Reporting summary.** Further information on research design is available in the Nature Research Reporting Summary linked to this article.

## Data availability
The source data underlying Figs. 1f–g, i, 2d–g, 3c–f, 4d and Supplementary Figs. 2b–c, 3a–b, 5b–c are provided as a Source Data file. All other data are available from the authors upon reasonable request.

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

## Acknowledgements

The authors would like to thank Jacob Kerssemakers and Federico Fanalista for help and fruitful discussions, and Eli van der Sluis for purification of FtsZ. This work was supported by the NWO TOP-PUNT grant (No. 718014001), the Netherlands Organisation for Scientific Research (NWO/OCW) as part of the NanoFront, Basyc and FOM (No. 110) programs, and European Research Council Advanced Grant SynDiv (No. 669598).

## Author contributions

S.D. and C.D. conceived the experiments. S.D., F.B., A.L., M.G.F.L., W.K.S., and S.W. performed the experiments. S.D., F.B., S.W., and L.R. analyzed the data. S.D., A.L., and C.D. wrote the paper, with input from the rest of the authors.

## Additional information

**Competing interests:** The authors declare no competing interests.

