## [Peer Review File · Nature Communications]

Reviewers' Comments:

Reviewer #1:

Remarks to the Author:

The authors report the use of a microfluidic-based platform to produce coacervates inside micron-sized liposomes. In a basic sense, this represents the formation of a higher order synthetic cell in a controllable fashion. The technical quality of the work is very good and the studies show some interesting results. The novel aspects of the work relate to the size of produced liposomes (which are very similar to the size of cells) and the use of alpha-hemolysin pores to trigger the coacervation within the liposome. In addition, the study of the coacervation process itself, is interesting. That said, the key ideas developed and expounded have been previously reported elsewhere. Most notably, Deng and Huck published a paper last year in *Angewandte Chemie*, which presents use of different (capillary-based) but related microfluidic structures to form of monodisperse coacervate organelles within liposomes. This work laid out the basic idea of constructing the same type of higher order synthetic cell, with the authors showing retention of DNA inside the coacervates, triggered release and *in vitro* transcription within coacervates. The experiments and technologies described in this 2017 paper greatly limit the novelty of the current work. For this reason, I would not recommend publication in *Nature Communications* and would suggest that authors look to develop the accompanying biological reactions before submission to a more appropriate journal, e.g. *ASC Synthetic Biology*. I should note that the authors did cite the Huck paper, but restrict their assessment of this work to the fact that selective permeability was not addressed.

Reviewer #2:

Remarks to the Author:

The paper by Deshpande et al. describes the use of microfluidics to generate liposomes containing a coacervate component. Pores are integrated into the liposomes to allow passive transport of secondary coacervate components to drive liquid-liquid phase separation. They then use this platform to include enzymes which drive the formation of coacervates inside the porous liposomes and include biomolecules and liposomes into the coacervate to demonstrate functionality. This is interesting work and could be interesting for the community only if further evidence was provided to strengthen the conclusions. Some conclusions they make have been previously reported and therefore the novelty should be highlighted or appropriate references included.

General comments:

The authors make some general conclusions about coacervate behaviour. For example. "FtsZ was present already before the phase separation occurred. This suggests that the spatial organization of sequestered molecules within the coacervate depends on the sequence and timing of the addition of parts."

"We measured a partition coefficient $PSUVs = 2.9 \pm 0.6$ ($n = 35$), indicating that the coacervates acted as an efficient lipid reservoir"

The partition coefficient was found to be $P_{\text{fluorescein}} = 2.4 \pm 0.4$ ($n = 31$). Summing up, we successfully demonstrated two important attributes of coacervates: sequestering and concentrating biomolecules, and serving as hubs for biochemical reactions.

These are all properties of coacervates which have been previously described. So are the authors describing these as new attributes? if so please state what is new about them? Or if they are confirming that these are well known properties of coacervates please include references to previous work which has described these properties.

The authors describe the use of the platform to "measure the time dependence of the coacervation process". They have extracted the fluorescence of the coacervate droplet during the nucleation and growth process and of the surrounding phase or the dilute phase. However, a complete description seems to be missing here. A number of systems have been used to demonstrate coacervate formation including pLL/ATP, RNA/Spermine and enzyme induced

formation of the RNA/spermine reactions but there are no real comparisons of the mechanism of formation despite saying in the discussion the different mixtures “differ in their nucleation behaviour”. There is a potentially a lot of interesting information here and new insights for the community. The authors allude to this but do not fully exploit the platform or the data they have obtained to make non-phenomenological conclusions. Especially as they say that “de novo nucleation of condensates in real time, a process which has been difficult to study and control so far” and describe how this system is able to overcome this. With regard to the analysis of the fluorescence intensity as function of time. In one instance, they describe the decrease in fluorescence intensity due to the droplet moving out of the field of view “most importantly, coacervates moving out of focus owing to the small depth of field ($< 1 \mu\text{m}$) of the objective” - why is this not the case in the other experiments? And how then can this data be used to make assertions about the time dependence of coacervation growth? Also, are there photobleaching effects as the SI videos seem to show photobleaching but this is not seen in the analysis. To get more information on the nucleation process would it be useful to get an analysis of size and number of droplets as function of time as this would offer kinetic information for the growth process. Indeed the authors describe a size dependence of the coacervates formed but its not clear where the data is to support their conclusions here.

More specific comments:

They describe 11% of their vesicles have non-specific membrane defects. It would be interesting to add some additional information about this. It is well known that polylysine will intercalate into lipid membranes increasing leakiness of membranes. If this were the reasons for the membrane leakiness why would only 11% have this effect. One could use fluorescent polylysine to see where it locates in the membrane and to see if it is intercalating.

The concentration of ATP and fluorescently labelled ATP and Polylysine should be included in the description of the experiments in the main text.

The authors state that the “finite amount of pLL molecules set the size of the formed coacervates”. Are there experiments which show varying size of the final coacervate droplet with different pLL concentrations to verify this statement?

“In a typical experiment, we measured a monodisperse population of liposomes ($d_{\text{vesicle}} = 14.2 \pm 1.0 \mu\text{m}$, $n = 213$) with monodisperse coacervates formed within them ($d_{\text{coacervates}} = 2.8 \pm 0.3 \mu\text{m}$, $n = 213$).” Was this with one set of experiment? The authors should verify how many experiments / repeats were undertaken.

“The results showed no accumulation of dextran molecules inside the coacervates but instead a homogenous fluorescence throughout the liposome, with a slight preference to reside at the coacervate interface (Supplementary Fig. 2). This observed accumulation at the interface clearly did not alter the coacervation process, as the coacervation dynamics remained unchanged, independent of whether we used sucrose or dextran to settle the liposomes”. Going back to point 2, this system could be analysed and modelled in the same way to observe any differences in nucleation-growth mechanisms when additional molecules are added.

A threshold ATP concentration is described to onset the coacervation process. What is this concentration? Additionally, they should make it clear that this coacervate formation observation is from observation of fluorescent ATP if this is the case.

“ATP molecules rapidly diffused throughout the well, and entered the liposomes through the pores” Is there a difference in the onset of coacervation with the addition of pores compared to the intrinsically porous membranes? The data should be available to make answer this point.

“We analyzed the time evolution of the total fluorescence intensity given by the bright pixels, i.e., those belonging to the condensed phase, in order to quantify the coacervation process.” A clearer description of how this is done is required. Is this from tracking? Is the code available?

“The value plateaued within a few minutes, with the coalescence of smaller coacervates into a single big coacervate affecting it negligibly.” Would this be affected by changing the concentration of ATP? Is there a connection here with the final size of droplets?

“More interestingly, we identified a short regime ($\sim 30 \text{ s}$), prior to the emergence of first coacervates, that showed a marked intensity decrease of the dilute phase. We speculate that this fluorescence intensity loss, just before the coacervates are observed, indicates the nucleation

process that precedes the observable formation of coacervates. The 20% decrease at the moment of appearance of visible structures would mean that about 20% of the pLL molecules had already nucleated before any visibly detectable coacervates." As it currently stands, this description is incredibly phenomenological and it would be of interest to the readership to make this more quantitative (See point 2). The observation about the 20% decrease can be attributed to the resolution of the microscope. How does this then affect the description of the nucleation-growth time course?

With regard to the apyrase experiment, the result that they observe i.e. that the coacervate droplet is conserved when apyrase is placed in the dispersion could also be attributed to the apyrase remaining on the outside of the lipid vesicle and not going through the pores. The current control experiment does not rule this out. Experiments should be shown to conclusively show that it is due to ATP transfer, for example, apyrase could be included into the vesicle with pLL and then ATP transferred into the inner lumen of the vesicle. The data which showed that without any pyranase the coacervation takes place in the same way supports the hypothesis that the pyranase is not penetrating into the interior of the lipid vesicles.

As described in point 2, analysis of formation of polyU/ Spermine coacervates in comparison to pLL/ ATP would be interesting.

"A time-series plot of the coacervate-phase intensity for the RNA/spermine system showed a definite but weak increase for an initial time period of about one minute, corresponding to the formation of multiple small coacervates (Fig. 3c, $n = 7$). However, the fluorescence intensity subsequently gradually decreased and remained stable at a value only slightly higher than that at time zero." This description is very phenomenological (see point 2 and others).

"The plot of the dilute-phase intensity showed a monotonous decrease and did not display any discontinuous transition as was seen in the case of pLL/ATP coacervation (Fig. 3d)." This could be clarified by changing the enzyme concentration and observing the kinetics of coacervate formation. Also, a comparison of polyU/ spermidine without enzyme and UDP/Spermine/PNPase kinetics would add to the current conclusions (see point 2 and others)

"With virtually unlimited supply of spermine and UDP, the finite reaction time of PNPase activity likely was the limiting parameter in the polymerization of RNA." Is there any analysis on the size of the droplets which are formed? Why would the coacervate not increase in size if there was continual exchange and the coacervate was allowed to continually grow. Some statement would be interesting here.

"The observed homogenous distribution contrasts the FtsZ localization at the interface of pLL/ATP coacervates that was reported recently, where FtsZ was added to a solution containing pre-formed, stable coacervates¹¹. In the current case, however, FtsZ was present already before the phase separation occurred. This suggests that the spatial organization of sequestered molecules within the coacervate depends on the sequence and timing of the addition of parts." Control experiments to be included in the SI showing this effect in coacervates with no vesicles are required to directly compare coacervates with vesicles and without.

"triggering pLL/ATP coacervation did concentrate the SUVs within the coacervates (Fig. 4b). We measured a partition coefficient $PSUVs = 2.9 \pm 0.6$ ($n = 35$), indicating that the coacervates acted as an efficient lipid reservoir. The SUVs were distributed homogeneously within the coacervate, as opposed to previously observed localization at the interface of pre-formed polyU/spermine coacervates" Was this because previous experiments added the SUVs to the dispersion of pre-made coacervates? This should be clarified with respect to previous experiments.

"We encapsulated β -galactosidase inside liposomes, which presumably partitioned into the coacervates once they formed, and then added FDG in the exterior environment." An experiment should be undertaken to prove that the FDG partitions into the coacervate phase, without this it is unclear whether increase in fluorescence is due to enzyme reaction taking place on the outside of the coacervate and fluorescein partitions in or the reaction is happening on the inside. Therefore, the following statement, I feel, is a little too strong for the experimental data. "Lastly, we showed a compartmentalized biochemical reaction, β -galactosidase converting its non-fluorescent substrate into fluorescein that remained confined within the coacervate."

"The fluorescence did not increase significantly over time, suggesting that the reaction quickly went to completion." If there was a constant flux of substrate why would this be the case?

In the discussion section, they say "we obtained temporal control over the onset of coacervation as well as the final size of condensates." Please make it clearer where the data is for this.

"coacervates inside liposomes. We observed that the condensation process could be roughly divided into two distinct regimes: an initial rapid phase separation followed by the coalescence of coacervates." The authors need more analysis of the data to back up this statement.

"as it was limited by the RNA polymerization rate. The coacervates did not grow indefinitely but soon reached a stable size," This was an interesting point, but further explanation with more experimental back up is required to make such a clear statement.

Again, more experimental back up or further analysis of the data is required to support the following statement. "A notable difference between the pLL/ATP and RNA/spermine systems was their condensation dynamics inside the liposome. While the pLL/ATP mixture underwent a spontaneous nucleation transition, condensates continuously grew in the RNA/spermine mixture. These observations suggest that both mixtures differ fundamentally in their nucleation behavior, and may be regarded as cases of homogeneous and heterogeneous nucleation, respectively³²."

In the discussion, the authors describe "While we used pLL/ATP coacervates to demonstrate these functions, we expect that other coacervate systems such as polyU/spermine will exhibit similar effects." It would not be very difficult to test the systems for the polyU/spermine system and include these into the paper.

Minor comments:

The authors should state the number of experiments in the figure legend.

The authors have only shown single droplets. Wider fields of view should be shown to be sure that this has not only happened in one vesicle and images of the statistics should be shown.

Point-by-point response to the reviewers' comments:

(Reviewers' comments are in blue font. Author responses are represented in italics and black font.)

Reviewer #1

The authors report the use of a microfluidic-based platform to produce coacervates inside micron-sized liposomes. In a basic sense, this represents the formation of a higher order synthetic cell in a controllable fashion. The technical quality of the work is very good and the studies show some interesting results. The novel aspects of the work relate to the size of produced liposomes (which are very similar to the size of cells) and the use of alpha-hemolysin pores to trigger the coacervation within the liposome. In addition, the study of the study of coacervation process itself, is interesting. That said, the key ideas developed and expounded have been previously be reported elsewhere. Most notably, Deng and Huck published a paper last year in *Angewandte Chemie*, which presents use of different (capillary-based) but related microfluidic structures to form of monodisperse coacervate organelles within liposomes. This work laid out the basic idea of constructing the same type of higher order synthetic cell, with the authors showing retention of DNA inside the coacervates, triggered release and in vitro transcription within coacervates. The experiments and technologies described in this 2017 paper greatly limit the novelty of the current work. For this reason, I would not recommend publication in *Nature Communications* and would suggest that authors look to develop the accompanying biological reactions before submission to a more appropriate journal, e.g. *ASC Synthetic Biology*. I should note that the authors did cite the Huck paper, but restrict their assessment of this work to the fact that selective permeability was not addressed.

We thank the reviewer for his/her appreciation of the high quality and interest of our work.

*We would like to point out that our work differs significantly from the previous work by Deng et al. (*Angew. Chemie*, 2017), on at least three major fronts:*

- (i) *Real-time coacervate dynamics – We externally induce and subsequently follow the dynamics of coacervation within individual liposomes, which was not achieved in the publication by Deng et al.*
- (ii) *External supply to induce coacervation – We use protein pores embedded in the membrane to transport a particular component into the vesicle, in order to induce coacervation in a controlled way. Such a strategy opens very interesting avenues, for example, to carry out step-by-step reaction sequences, something that is not possible in the closed system reported by Deng et al.*
- (iii) *Use of cell-sized containers – We use cell-sized (10 µm) liposomes as compared to 100 µm diameter vesicles used by Deng et al.*

For these reasons, we do not think that the published work by Deng et al. limits the novelty of our work in a significant way.

Reviewer #2

The paper by Deshpande at al. describes the use of microfluidics to generate liposomes containing a coacervate component. Pores are integrated into the liposomes to allow passive transport of secondary coacervate components to drive liquid-liquid phase separation. They then use this platform to include enzymes which drive the formation of coacervates inside the porous liposomes and include biomolecules and liposomes into the coacervate to demonstrate functionality. This is interesting work and could be interesting for the community only if further evidence was provided to strengthen the conclusions. Some conclusions they make have been previously reported and therefore the novelty should be highlighted or appropriate references included.

We thank the reviewer for the appreciation of the interest of our work. Below we address his/her comments:

General comments:

The authors make some general conclusions about coacervate behaviour. For example. "FtsZ was present already before the phase separation occurred. This suggests that the spatial organization of sequestered molecules within the coacervate depends on the sequence and timing of the addition of parts."

“We measured a partition coefficient $PSUVs = 2.9 \pm 0.6$ ($n = 35$), indicating that the coacervates acted as an efficient lipid reservoir”

The partition coefficient was found to be $P_{\text{fluorescein}} = 2.4 \pm 0.4$ ($n = 31$). Summing up, we successfully demonstrated two important attributes of coacervates: sequestering and concentrating biomolecules, and serving as hubs for biochemical reactions.

These are all properties of coacervates which have been previously described. So are the authors describing these as new attributes? if so please state what is new about them? Or if they are confirming that these are well known properties of coacervates please include references to previous work which has described these properties.

We are of course aware that the properties of coacervates that we showed have been described previously. Our intention was not at all to describe them as new attributes, but to confirm their behavior in our coacervate-in-liposome system. We have now included the appropriate references and changed the wording slightly to avoid this confusion, see page 12.

The authors describe the use of the platform to “measure the time dependence of the coacervation process”. They have extracted the fluorescence of the coacervate droplet during the nucleation and growth process and of the surrounding phase or the dilute phase. However, a complete description seems to be missing here. A number of systems have been used to demonstrate coacervate formation including pLL/ATP, RNA/Spermine and enzyme induced formation of the RNA/spermine reactions but there are no real comparisons of the mechanism of formation despite saying in the discussion the different mixtures “differ in their nucleation behaviour”. There is a potentially a lot of interesting information here and new insights for the community. The authors allude to this but do not fully exploit the platform or the data they have obtained to make non-phenomenological conclusions. Especially as they say that “de novo nucleation of condensates in real time, a process which has been difficult to study and control so far” and describe how this system is able to overcome this.

We thank the reviewer for this comment. The main aim of our manuscript was to demonstrate a novel system to study coacervation processes that occur at a similar scale as that of living cells. Using two discrete coacervate systems, i.e., pLL/ATP and RNA/spermine, we showed the proof-of-principle. While the basic mechanism for these two coacervation processes is the same (electrostatic interaction between oppositely charged polyelectrolytes), we found some marked difference in their dynamics. We have now performed more detailed analyses where we examined the time evolution of the number and size of the coacervates (Fig. 2f-g, Fig 3e-f). We have now also emphasized the differences between the two systems in the results and discussion. We have also significantly expanded the discussion section to indicate the benefits and possible uses of our system, see page 15.

With regard to the analysis of the fluorescence intensity as function of time. In one instance, they describe the decrease in fluorescence intensity due to the droplet moving out of the field of view “most importantly, coacervates moving out of focus owing to the small depth of field ($< 1 \mu\text{m}$) of the objective”- why is this not the case in the other experiments? And how then can this data be used to make assertions about the time dependence of coacervation growth?

We thank the reviewer for this comment that prompted us to analyze the data more carefully. Upon re-evaluating a larger set of data, we observe that the intensity does not decay but instead stays roughly constant, albeit with strong fluctuations. We have modified the text on page 11.

Also, are there photobleaching effects as the SI videos seem to show photobleaching but this is not seen in the analysis.

We can safely exclude any photobleaching effects. Liposomes within the same experiment that did not form coacervates act as an ideal control to test photobleaching. We did not see any decrease in the fluorescence, both for pLL/ATP and polyU/spermine. We have now included these data as Supplementary Figure 3. We have also clarified these points in the text on page 9 and 12.

The sudden decrease seen at the end of Supplementary Video 5 was due to a technical error during acquisition and has not affected our analyses. To avoid any confusion, we have now removed the very last part of this video.

To get more information on the nucleation process would it be useful to get an analysis of size and number of droplets as function of time as this would offer kinetic information for the growth process. Indeed the authors describe a size dependence of the coacervates formed but its not clear where the data is to support their conclusions here.

We would like to thank the reviewer for his/her suggestions. We have now performed a more detailed analysis regarding the size and the number of coacervates as a function of time for both the systems, and have included it in the main text on page 8 and 11, as well as in Fig. 2f-g and Fig. 3e-f.

More specific comments:

They describe 11% of their vesicles have non-specific membrane defects. It would be interesting to add some additional information about this. It is well known that polylysine will intercalate into lipid membranes increasing leakiness of membranes. If this were the reasons for the membrane leakiness why would only 11% have this effect. One could use fluorescent polylysine to see where it locates in the membrane and to see if it is intercalating.

We indeed used fluorescently labelled polylysine in our experiments to follow the coacervation process. We did not see any noticeable intercalation in the membrane, which would appear in the form of an increased fluorescence intensity at the membrane. We thus conclude that some of the liposomes developed non-specific membrane defects or experienced transient pore formation-resealing events, which have been well documented in the literature. We have now included these references in the manuscript (page 6).

The concentration of ATP and fluorescently labelled ATP and Polylysine should be included in the description of the experiments in the main text.

We have now included the concentrations of the coacervate components in the main text.

The authors state that the “finite amount of pLL molecules set the size of the formed coacervates”. Are there experiments which show varying size of the final coacervate droplet with different pLL concentrations to verify this statement?

We expect the coacervate size to be limited by the encapsulated component, since the other component is present externally in a near infinite amount. We have now included experiments with varying amount of pLL (1.2 mg/mL, 5.5 mg/mL, and 15.2 mg/mL), that indeed show this is the case (Fig. 1g-h), i.e., the size of the coacervate increases with the amount of pLL encapsulated in the liposome. We have also included a plot showing that the coacervate volume scales linearly with the pLL concentration, see Fig. 1i. We now also clarified this point in the text on page 7.

“In a typical experiment, we measured a monodisperse population of liposomes (dvesicle = $14.2 \pm 1.0 \mu\text{m}$, n = 213) with monodisperse coacervates formed within them (dcoacervates = $2.8 \pm 0.3 \mu\text{m}$, n = 213).” Was this with one set of experiment? The authors should verify how many experiments / repeats were undertaken.

The monodispersity was indeed analyzed for one set of experiment and this information has now been included more explicitly in the main text on page 7.

“The results showed no accumulation of dextran molecules inside the coacervates but instead a homogenous fluorescence throughout the liposome, with a slight preference to reside at the coacervate interface (Supplementary Fig. 2). This observed accumulation at the interface clearly did not alter the coacervation process, as the coacervation dynamics remained unchanged, independent of whether we used sucrose or dextran to settle the liposomes”. Going back to point 2, this system could be analysed and modelled in the same way to observe any differences in nucleation-growth mechanisms when additional molecules are added.

We did not observe any significant effect of additional molecules on the coacervate dynamics. In order to settle the liposomes at the bottom of the well, we needed to encapsulate a small amount of high-density substance inside liposomes. We chose sucrose and dextran due to their inert nature and relatively high density. We did not see any difference in the coacervation dynamics when using sucrose (Fig. 2) or dextran (Supplementary Fig. 4). We feel that investigating the particular effect of additional molecules is beyond the scope of this paper.

A threshold ATP concentration is described to onset the coacervation process. What is this concentration? Additionally, they should make it clear that this coacervate formation observation is from observation of fluorescent ATP if this is the case.

We have now included a turbidity assay that we used to determine the threshold ATP concentration of about 6 mM (Fig. 1f and the text on page 7). The coacervate formation is observed using the fluorescence from FITC-pLL and this is now mentioned in the figure caption as well as in the main text on page 8.

“ATP molecules rapidly diffused throughout the well, and entered the liposomes through the pores” Is there a difference in the onset of coacervation with the addition of pores compared to the intrinsically porous membranes? The data should be available to make answer this point.

We saw no difference in the onset of coacervation between the liposome containing membrane pores and those with an inherently leaky membrane. We have now included Supplementary Fig. 6 showing a time-lapse of a coacervation event within an intrinsically porous liposome.

“We analyzed the time evolution of the total fluorescence intensity given by the bright pixels, i.e., those belonging to the condensed phase, in order to quantify the coacervation process.” A clearer description of how this is done is required. Is this from tracking? Is the code available?

We have now described the image analysis in much greater detail in the Methods section, explaining the way that the coacervate-phase and dilute-phase intensities were measured and the way that the number and size of coacervates were calculated, see page 8 and 23.

“The value plateaued within a few minutes, with the coalescence of smaller coacervates into a single big coacervate affecting it negligibly.” Would this be affected by changing the concentration of ATP? Is there a connection here with the final size of droplets?

We have now backed up our observation with a new analysis regarding the number and the size of the coacervates (page 9). Since the coacervate size was set by the amount of pLL encapsulated within the liposome, it is not affected by changing the ATP concentration (as long as it above the threshold concentration of 6 mM).

“More interestingly, we identified a short regime (~30 s), prior to the emergence of first coacervates, that showed a marked intensity decrease of the dilute phase. We speculate that this fluorescence intensity loss, just before the coacervates are observed, indicates the nucleation process that precedes the observable formation of coacervates. The 20% decrease at the moment of appearance of visible structures would mean that about 20% of the pLL molecules had already nucleated before any visibly detectable coacervates.” As it currently stands, this description is incredibly phenomenological and it would be of interest to the readership to make this more quantitative (See point 2). The observation about the 20% decrease can be attributed to the resolution of the microscope. How does this then affect the description of the nucleation-growth time course?

We have now included a more detailed analysis and comparison between the two systems, as suggested by the reviewer (see also his/her earlier comments). Quantitatively, it is hard to go beyond the rough estimation of 20%.

With regard to the apyrase experiment, the result that they observe i.e. that the coacervate droplet is conserved when apyrase is placed in the dispersion could also be attributed to the apyrase remaining on the outside of the lipid vesicle and not going through the pores. The current control experiment does not rule this out. Experiments should be shown to conclusively show that it is due to ATP transfer, for example, apyrase could be included into the vesicle with pLL and then ATP transferred into the inner lumen of the vesicle. The data which

showed that without any pyranase the coacervation takes place in the same way supports the hypothesis that the pyranase is not penetrating into the interior of the lipid vesicles.

We completely agree with the reviewer and would like to clarify a miscommunication regarding this point. Apyrase was indeed encapsulated inside the vesicle, exactly as suggested by the reviewer. We have now phrased this more explicitly in the text on page 9.

As described in point 2, analysis of formation of polyU/ Spermine coacervates in comparison to pLL/ ATP would be interesting.

“A time-series plot of the coacervate-phase intensity for the RNA/spermine system showed a definite but weak increase for an initial time period of about one minute, corresponding to the formation of multiple small coacervates (Fig. 3c, n = 7). However, the fluorescence intensity subsequently gradually decreased and remained stable at a value only slightly higher than that at time zero.’ This description is very phenomenological (see point 2 and others).

“The plot of the dilute-phase intensity showed a monotonous decrease and did not display any discontinuous transition as was seen in the case of pLL/ATP coacervation (Fig. 3d).” This could be clarified by changing the enzyme concentration and observing the kinetics of coacervate formation. Also, a comparison of polyU/spermidine without enzyme and UDP/Spermine/PNPase kinetics would add to the current conclusions (see point 2 and others)

We would like to thank the reviewer for pointing out the missing detailed comparison between the two systems that we have studied. We have now revisited the analysis, improved on the explanation, and extensively compare the two systems in the Results (page 11) and Discussion section (page 17).

“With virtually unlimited supply of spermine and UDP, the finite reaction time of PNPase activity likely was the limiting parameter in the polymerization of RNA.” Is there any analysis on the size of the droplets which are formed? Why would the coacervate not increase in size if there was continual exchange and the coacervate was allowed to continually grow. Some statement would be interesting here.

We agree with the reviewer that, given unlimited supply of raw material, one would expect the coacervate to grow continuously. However, this is not what we observe. In fact, the coacervate reached its final size already within the first 10 minutes, which can be now seen from the analysis that we have carried out regarding the size of the polyU/spermine coacervates as a function of time. We have updated the text on this (page 12) and also included a Supplementary Fig. 7, showing polyU/spermine coacervates within liposomes, 2 hours after the production.

“The observed homogenous distribution contrasts the FtsZ localization at the interface of pLL/ATP coacervates that was reported recently, where FtsZ was added to a solution containing pre-formed, stable coacervates¹¹. In the current case, however, FtsZ was present already before the phase separation occurred. This suggests that the spatial organization of sequestered molecules within the coacervate depends on the sequence and timing of the addition of parts.” Control experiments to be included in the SI showing this effect in coacervates with no vesicles are required to directly compare coacervates with vesicles and without.

We have now conducted bulk control experiments without involving liposomes that show a similar sequestration of FtsZ in the coacervates (Supplementary Fig. 8), which we now mention on page 13.

“triggering pLL/ATP coacervation did concentrate the SUVs within the coacervates (Fig. 4b). We measured a partition coefficient $PSUVs = 2.9 \pm 0.6$ (n = 35), indicating that the coacervates acted as an efficient lipid reservoir. The SUVs were distributed homogeneously within the coacervate, as opposed to previously observed localization at the interface of pre-formed polyU/spermine coacervates” Was this because previous experiments added the SUVs to the dispersion of pre-made coacervates? This should be clarified with respect to previous experiments.

We thank the reviewer for prompting us to clarify our statement. Indeed, in the previously reported experiments, SUVs were added to pre-formed polyU/spermine coacervates. Additionally, we have now also conducted a bulk

control experiment that shows sequestration of SUVs in the coacervates also in absence of liposomes (Supplementary Fig. 9). We have clarified the text on page 13.

“We encapsulated β -galactosidase inside liposomes, which presumably partitioned into the coacervates once they formed, and then added FDG in the exterior environment.” An experiment should be undertaken to prove that the FDG partitions into the coacervate phase, without this it is unclear whether increase in fluorescence is due to enzyme reaction taking place on the outside of the coacervate and fluorescein partitions in or the reaction is happening on the inside. Therefore, the following statement, I feel, is a little too strong for the experimental data. “Lastly, we showed a compartmentalized biochemical reaction, β -galactosidase converting its non-fluorescent substrate into fluorescein that remained confined within the coacervate.”

We thank the reviewer for raising this valid point. In order to discriminate whether β -galactosidase partitions into the coacervate or not, we have now conducted a bulk experiment, in the absence of liposomes (Supplementary Fig. 10). We used the same experimental conditions to make the coacervates in the presence of the enzyme and then added the substrate (FDG). While we observed the fluorescence (emitted by the product fluorescein) getting emitted from both the coacervates as well as the surrounding dilute phase, the signal was clearly higher from the coacervate phase throughout the time-lapse. This demonstrates that indeed β -galactosidase is partitioned more into the coacervate, leading to a local increase in the fluorescence concentration. If the enzyme was primarily residing in the dilute phase, we would instead have seen the background intensity to increase. We now mention these experiments on page 14.

“The fluorescence did not increase significantly over time, suggesting that the reaction quickly went to completion.” If there was a constant flux of substrate why would this be the case?

We thank the reviewer for pointing this out. We have repeated the experiment and have modified Fig. 4c and the text on page 14 accordingly.

In the discussion section, they say “we obtained temporal control over the onset of coacervation as well as the final size of condensates.” Please make it clearer where the data is for this.

With our approach, we obtain temporal control because coacervation is not induced until the other component (e.g. ATP) is added to the collection well. The size of the condensate is determined by the pLL concentration, and we have now conducted additional experiments (Fig. 1g-i) as described earlier. We have now clarified this point by adding text to the Discussion section (page 15).

“coacervates inside liposomes. We observed that the condensation process could be roughly divided into two distinct regimes: an initial rapid phase separation followed by the coalescence of coacervates.” The authors need more analysis of the data to back up this statement.

We have now performed further analysis on the evolution of coacervates over time in terms of their size and number (Fig. 2f-g). This analysis clearly shows that numerous small coacervates are formed initially (in the first minute), which then fuse with each other to form a single large coacervate. To describe this, we updated and significantly expanded the text on page 8 and 16.

“as it was limited by the RNA polymerization rate. The coacervates did not grow indefinitely but soon reached a stable size,” This was an interesting point, but further explanation with more experimental back up is required to make such a clear statement.

We agree with the reviewer that this is an interesting observation. One plausible explanation could be the exoribonuclease activity of the enzyme, which would degrade the long polyU RNA into progressively shorter fragments making them unsuitable for coacervation. We have now added a discussion and experimental data in the Results and Discussion section, see page 12 and 17.

Again, more experimental back up or further analysis of the data is required to support the following statement. “A notable difference between the pLL/ATP and RNA/spermine systems was their condensation dynamics inside the liposome. While the pLL/ATP mixture underwent a spontaneous nucleation transition, condensates

continuously grew in the RNA/spermine mixture. These observations suggest that both mixtures differ fundamentally in their nucleation behavior, and may be regarded as cases of homogeneous and heterogeneous nucleation, respectively³².”

We have now performed more detailed analyses by measuring the size and the number of condensates over time (Fig. 2f-g and Fig. 3e-f). What we find is that coacervation for pLL/ATP starts simultaneously at several places suggesting homogeneous nucleation. Over time, these coacervates grow steadily, fuse with each other, and ultimately form a single, large coacervate. In case of polyU/spermine, the scenario is significantly different. Coacervation does start at multiple, but relatively fewer spots, probably where the polyU concentration is high enough to commence coacervation, suggesting heterogeneous nucleation. These coacervates do not grow much in size even after a few hours, as compared to the pLL/ATP coacervates. We have now updated the text in the results and discussion section, see page 8, 11 and 17.

In the discussion, the authors describe “While we used pLL/ATP coacervates to demonstrate these functions, we expect that other coacervate systems such as polyU/spermine will exhibit similar effects.” It would not be very difficult to test the systems for the polyU/spermine system and include these into the paper.

The main purpose of FtsZ sequestration, SUV sequestration, and compartmentalization of the β -galactosidase reaction was to show the broad utility of the coacervate-in-liposome system that we have developed. We chose the pLL/ATP system because of its relative simplicity compared to the polyU/spermine system, which involves an additional enzymatic reaction. While doing similar experiments in the polyU/spermine and other systems is potentially feasible, we feel that it would not add significant value to the paper. Nevertheless, in order to avoid unnecessary claims, we have now removed that specific sentence from the manuscript.

Minor comments:

The authors should state the number of experiments in the figure legend.

We have now included the number of experiments, in the figure legend, where appropriate.

The authors have only shown single droplets. Wider fields of view should be shown to be sure that this has not only happened in one vesicle and images of the statistics should be shown.

We have focused on single liposomes for Fig. 2 and Fig. 3 in order to allow for the best visualization of the coacervation process. A wider field-of-view in case of pLL/ATP coacervation is shown in Fig. 1d. We have now also included a wider field-of-view for the polyU/spermine coacervation as a Supplementary Figure 7. Furthermore, we have now calculated the efficiency of coacervate formation inside the liposomes and have included this information in the main text on page 6.

We sincerely would like to thank the reviewer for his/her very thorough reading and for the many valuable suggestions for improvements to the manuscript. We hope that we have satisfactorily addressed all the issues.

Reviewers' Comments:

Reviewer #2:

Remarks to the Author:

Manuscript NCOMMS-18-28789A

The manuscript has been much improved, however there are still some things which are unclear.

1. In figures 1e, 2b, supplementary figures 1c and 7 (and this is very evident in supplementary figure 7) that there are coacervates forming on the outside. If the liposomes are encapsulating the coacervate components as stated why would you see evidence of coacervates on the outside of the vesicle. Additionally, does the polyU that is being produced in the coacervate also diffuse out of the coacervate as it is being formed?
2. With regard to the image analysis that is being undertaken to obtain the kinetics of nucleation-growth of the coacervate within the liposome. It is still not entirely clear whether these images are wide-field images or confocal images and therefore if the total fluorescence intensity is truly accounted for. These really need 3d stacks to obtain the total intensity over time. Especially as the images appear to show out of focus fluorescent coacervate droplets within the liposomes. How are these accounted for in the image analysis?
3. The authors have included a more thorough description of the differences between polyU/spermine coacervae formation and PLL/ATP coacervate formation. The authors should additionally provide and comment on the number of vesicles which show successful coacervate formation.
4. The authors also state that the dextran does not go into the coacervate droplet, however, it is not clear how they can make this ascertainment by eye. Also whether the image has been obtained on widefield or confocal should make a difference. Quantitative analysis of appropriate imaging would be required to make this ascertainment.
5. It does not seem obvious as to why the ATP would penetrate lipid vesicles which still contain have the octanol droplet. Is ATP even soluble in octanol? How would this happen?
6. Lines 131- 134, the authors give numbers for the probability of forming a hybrid container but there is no experimental evidence for this anywhere.

Point-by-point response to the reviewers' comments:

(Reviewers' comments are in blue font. Author responses are represented in italics and black font.)

Reviewer #2

The manuscript has been much improved, however there are still some things which are unclear.

We thank the reviewer for acknowledging our efforts in improving the manuscript according to his/her suggestions.

1. In figures 1e, 2b, supplementary figures 1c and 7 (and this is very evident in supplementary figure 7) that there are coacervates forming on the outside. If the liposomes are encapsulating the coacervate components as stated why would you see evidence of coacervates on the outside of the vesicle. Additionally, does the polyU that is being produced in the coacervate also diffuse out of the coacervate as it is being formed?

We thank the reviewer for pointing out this observation. The occasional presence of coacervates outside liposomes is due to the undesired release of some amount of coacervate components (e.g. pLL) associated with the liposome production. We have now clarified this on page 7.

On the second question: polyU has a very low probability of diffusing out of the coacervate as it is being formed, due to its increasingly large size and it being associated with the enzyme PNPase.

2. With regard to the image analysis that is being undertaken to obtain the kinetics of nucleation-growth of the coacervate within the liposome. It is still not entirely clear whether these images are wide-field images or confocal images and therefore if the total fluorescence intensity is truly accounted for. These really need 3d stacks to obtain the total intensity over time. Especially as the images appear to show out of focus fluorescent coacervate droplets within the liposomes. How are these accounted for in the image analysis?

We are happy to further clarify the details of our imaging technique. We obtained and analyzed wide-field fluorescence images, not confocal images. In order to remove any ambiguity, we now mention it explicitly on page 23.

As the coacervation process results in freely diffusing coacervate particles, using wide-field fluorescence microscopy is advantageous as it allows, at a moderately fast rate, to acquire the intensity across a thick region of the liposome for two different fluorescence channels (one to detect the liposome and the other one to detect coacervation). We are aware that the obtained intensities do not represent the total fluorescence intensity of the entire liposome, and we have now clarified this on page 24.

However, we note that our approach allows us to analyze the coacervation dynamics within the liposomes. As the reviewer has suggested, 3D confocal imaging can be used to obtain the total fluorescence intensity of the entire liposome. However, this may be challenging as it needs very high-speed sampling, since the formed coacervates are free to diffuse within the liposome and will move significantly during data acquisition (for example, a 1 μm diameter coacervate diffuses $\sim 1.6 \mu\text{m}$ in 1 s).

The out-of-focus coacervates usually have their fluorescence intensity much below the threshold value and thus do not contribute to the coacervate phase intensity. In case of the evolution of coacervate size over time, only in-focus coacervates were measured, as stated on page 23.

3. The authors have included a more thorough description of the differences between polyU/spermine coacervae formation and PLL/ATP coacervate formation. The authors should additionally provide and comment on the number of vesicles which show successful coacervate formation.

We have now included the number of vesicles showing successful coacervate formation for the pLL/ATP (page 8) as well as for polyU/spermine system (page 11).

4. The authors also state that the dextran does not go into the coacervate droplet, however, it is not clear how they can make this ascertainment by eye. Also whether the image has been obtained on widefield or confocal

should make a difference. Quantitative analysis of appropriate imaging would be required to make this ascertainment.

We thank the reviewer for prompting us to do a quantitative analysis on the data to back our claim that dextran does not accumulate in the coacervate. We have now included a bar graph showing a similar fluorescence intensity of AF647-dextran inside the coacervate and in the liposome lumen (Supplementary Fig. 2b). We have also included an average line profile across the coacervates, that shows a slightly increased fluorescence intensity of AF647-dextran at the interface (Supplementary Fig. 2c). These analyses further support and merit our statement (on page 7, now slightly modified).

5. It does not seem obvious as to why the ATP would penetrate lipid vesicles which still contain have the octanol droplet. Is ATP even soluble in octanol? How would this happen?

We agree with the reviewer that this is non-obvious. As ATP is not soluble in 1-octanol, it is certainly not diffusing across the 1-octanol droplet. As we stated in the text (page 6), we speculate that ATP is able to pass through the interfacial region between the liposome and the 1-octanol pocket, which potentially is in a disordered state as the bilayer is still being formed. Although potentially of some interest, we feel that further experimentation to pinpoint the exact cause of this phenomenon is beyond the scope of current paper.

6. Lines 131- 134, the authors give numbers for the probability of forming a hybrid container but there is no experimental evidence for this anywhere.

We now include the statistics that we used to calculate the probability of forming a hybrid container in the Methods section (page 23).

Finally, we would again like to thank the reviewer for his/her additional suggestions that has further improved the manuscript. We hope that we have now satisfactorily addressed all the issues.

Reviewers' Comments:

Reviewer #2:

Remarks to the Author:

Dear authors,

all queries put forward has been addressed. Thank you for your efforts. However, as there is still a little uncertainty of the value of the analysis of droplet formation using wide field images, please ensure that the methodology is well understood by referencing the additional explanations from the materials and methods/SI in the main text and the figure legend.

It still seems strange to me that ATP will go through the lipid and octane pocket (point 5). The authors have said that the experiments are beyond the scope of the paper therefore, please add a reference to back up your statement.

Reviewer #3:

Remarks to the Author:

Summary:

The manuscript "Spatiotemporal control of coacervate formation within liposomes" describes a creative and efficient approach to spatially and temporally control liquid-liquid phase separation (LLPS). The authors attempt to demonstrate the power of this approach, by showing not only that they can design and control LLPS conditions, but that they can further resolve differences in nucleation dynamics of distinct coacervate systems. It is this particular result – and more specifically, the method/analysis used to arrive at this result, that appears to cause Reviewer 2 much concern. Here, as instructed, comments addressing these concerns only, are summarized. Overall, I do agree with the concerns of Reviewer 2, and offer suggestions for relatively minor revisions in analysis/data representation that should placate these concerns, while still obviating the need for new/additional measurements.

Comments:

Using widefield imaging in order to capture a representative feature of the overall behavior within the liposome sphere should in principle, be a legitimate approach -- so long as the authors don't attempt to quantify values that 1) rely on total intensities within a liposome or 2) are subject to significant differences for in v.s out of focus objects. While the authors suggest in their rebuttal that they are not attempting to make such quantifications, the results – in particular the data on the 'coacervate phase' quantification do still suffer from artifacts related to out of focus objects.

In order to quantify the coacervate phase, the authors are summing the intensities of the pixels attributed to the phase (initially defined by a threshold value). This is very problematic using wide-field microscopy, as both the pixel areas of the droplets and their relative brightness will deviate significantly for droplets at varying degrees of focus. However, the authors don't necessarily need to make these quantifications to make their point. Their raw images appear to demonstrate distinct assembly mechanisms in 2 LLPS systems. In the first (Fig2), many droplets arise quickly – concomitant with a rapid depletion of background intensity. In the second (Fig3), the background decreases much more steadily, with the onset of a smaller numbers of droplets. These apparent differences could still be captured by comparing the average background intensity of the droplets, along with the droplet number in the representative field. (Though it is worth noting that the average intensity of the background could still be influenced by smaller out of focus droplets – and it would be a good idea to still discuss that possibility in the text).

Based on the above comments, I would suggest the following minor revisions:

- Do not include the 'coacervate phase' panels (Fig 2d, Fig3c).

- Replace the 'dilute phase' data (Fig 2e and 3d) with average intensity of the background (probably more appropriately done by averaging the intensity over the number of pixels below the threshold instead of simply adding them). Also, include discussion in the text, that the presence of small out of focus droplets, could still contribute to these values.
- Keep the '# of coacervates' panel as is (Fig 2f, Fig3e)
- If keeping the 'coacervate diameter' panel – text must include clear statement that there is significant error in these values due to out of focus objects.
- Finally, Please align all the graphs to the same axis! In other words, have the '# of coacervate' graph have a zero baseline before jumping at t_0 .

Point-by-point response to the reviewers' comments:

(Reviewers' comments are in blue font. Author responses are represented in italics and black font.)

Reviewer #2

All queries put forward has been addressed. Thank you for your efforts. However, as there is still a little uncertainty of the value of the analysis of droplet formation using wide field images, please ensure that the methodology is well understood by referencing the additional explanations from the materials and methods/SI in the main text and the figure legend.

We thank the reviewer for acknowledging our efforts. We have now made sure that the methodology is clear by further referencing the additional explanation from the Methods section in the main text and the figure legends (pages 11, 24, 35, 37).

It still seems strange to me that ATP will go through the lipid and octane pocket (point 5). The authors have said that the experiments are beyond the scope of the paper therefore, please add a reference to back up your statement.

We have now added appropriate references to back up our statement (page 6).

Reviewer #3

The manuscript "Spatiotemporal control of coacervate formation within liposomes" describes a creative and efficient approach to spatially and temporally control liquid-liquid phase separation (LLPS). The authors attempt to demonstrate the power of this approach, by showing not only that they can design and control LLPS conditions, but that they can further resolve differences in nucleation dynamics of distinct coacervate systems. It is this particular result – and more specifically, the method/analysis used to arrive at this result, that appears to cause Reviewer 2 much concern. Here, as instructed, comments addressing these concerns only, are summarized. Overall, I do agree with the concerns of Reviewer 2, and offer suggestions for relatively minor revisions in analysis/data representation that should placate these concerns, while still obviating the need for new/additional measurements.

We thank the reviewer for succinctly summarizing our work. We are aware of possible drawbacks of using wide-field microscopy. We have been very clear about the limitations of our analysis and have not made any unnecessary claims. We thank the referee for the suggestions for minor revisions, which we have largely adopted as described below.

Using widefield imaging in order to capture a representative feature of the overall behavior within the liposome sphere should in principle, be a legitimate approach -- so long as the authors don't attempt to quantify values that 1) rely on total intensities within a liposome or 2) are subject to significant differences for in v.s out of focus objects. While the authors suggest in their rebuttal that they are not attempting to make such quantifications, the results – in particular the data on the 'coacervate phase' quantification do still suffer from artifacts related to out of focus objects.

We agree with the referee. The signal due to out-of-focus objects acts like a weak background noise and increases the error on the trendline. Although the quantification of the coacervate phase suffers from out-of-focus objects to some extent, it still, however, allows us to clearly indicate the emergence of the liquid-liquid phase separation inside liposomes, which is the main focus of the paper.

In order to quantify the coacervate phase, the authors are summing the intensities of the pixels attributed to the phase (initially defined by a threshold value). This is very problematic using wide-field microscopy, as both the pixel areas of the droplets and their relative brightness will deviate significantly for droplets at varying degrees of focus. However, the authors don't necessarily need to make these

quantifications to make their point. Their raw images appear to demonstrate distinct assembly mechanisms in 2 LLPS systems. In the first (Fig2), many droplets arise quickly – concomitant with a rapid depletion of background intensity. In the second (Fig3), the background decreases much more steadily, with the onset of a smaller numbers of droplets. These apparent differences could still be captured by comparing the average background intensity of the droplets, along with the droplet number in the representative field. (Though it is worth noting that the average intensity of the background could still be influenced by smaller out of focus droplets – and it would be a good idea to still discuss that possibility in the text).

Naturally, we agree with the reviewer that the fluorescence intensity will depend on whether the object is in focus or not. Although out-of-focus coacervates usually have their fluorescence intensity much below the threshold value, they could still contribute to the measured value of the background to some extent. We are also aware that the obtained intensities do not represent the total fluorescence intensity of the entire liposome, but rather a section of it. However, as also acknowledged by the referee, our approach clearly allows us to qualitatively analyze the coacervation dynamics within liposomes.

We now added a comment on this point to the Methods section (page 24).

Based on the above comments, I would suggest the following minor revisions:

- Do not include the 'coacervate phase' panels (Fig 2d, Fig3c).

We feel that these panels are actually quite beneficial to the reader to understand the processes that we are describing, and thus we strongly prefer to keep them. At the same time, we realize their more qualitative nature and therefore we have now switched the order and emphasis: We now first show the quantitative analysis (size and diameter) and only then the time dependence. We have further described the analysis in detail in the Methods section, along with its limitations.

- Replace the 'dilute phase' data (Fig 2e and 3d) with average intensity of the background (probably more appropriately done by averaging the intensity over the number of pixels below the threshold instead of simply adding them). Also, include discussion in the text, that the presence of small out of focus droplets, could still contribute to these values.

We thank the reviewer for this suggestion. Accordingly, we have divided the dilute phase by the liposome area, in order to obtain a measure for a change in the amount of dilute phase over time (page 24).

- Keep the '# of coacervates' panel as is (Fig 2f, Fig3e)

We agree and have not removed these panels.

- If keeping the 'coacervate diameter' panel – text must include clear statement that there is significant error in these values due to out of focus objects.

For measuring the coacervate diameter, we have strictly measured only the in-focus objects. This was specifically done in order to avoid erroneous values. We are now mentioning this explicitly on page 24.

- Finally, Please align all the graphs to the same axis! In other words, have the '# of coacervate' graph have a zero baseline before jumping at t0.

We thank the reviewer for this useful suggestion. We have now modified the graphs accordingly (Fig. 2f-g and Fig. 3e-f).

Finally, we would like to thank all three reviewers for their helpful suggestions that have improved the manuscript. We trust that we have now satisfactorily addressed all the issues.